# Exceptional high AOD over Svalbard in Summer 2019: A multi-instrumental approach

Sara Herrero-Anta[1, 2], Sabine Eckhardt[3], Nikolaos Evangeliou[3], Stefania Gilardoni[4], Sandra Graßl[5, 6], Dominic Heslin-Rees[7], Stelios Kazadzis[8], Natalia Kouremeti[8], Radovan Krejci[7], David Mateos[1, 2], Mauro Mazzola[4], Christoph Ritter[6], Roberto Román[1, 2], Kerstin Stebel[3], and Tymon Zielinski[9]

[1]Group of Atmospheric Optics, Universidad de Valladolid, Valladolid, 47011, Spain
[2]Laboratory of Disruptive Interdisciplinary Science (LaDIS), Universidad de Valladolid, Valladolid, 47011, Spain
[3]NILU, Kjeller, 2007, Norway
[4]National Research Council, Institute of Polar Science (CNR-ISP), Bologna, 40129, Italy
[5]Institute of Physics and Astronomy, University of Potsdam, Potsdam, 14476, Germany
[6]Alfred-Wegener-Institute Helmholtz Centre for Polar and Marine Research, Potsdam, 14473, Germany
[7]Department of of Environmental Science, Stockholm University, Stockholm, S10691, Sweden
[8]Physikalisch-Meteorologisches Observatorium Davos/World Radiation Centre (PMOD/WRC), Davos, 7260, Switzerland
[9]Polish Academy of Sciences, Institute of Oceanology, Sopot, 81-712, Poland

**Correspondence:** Sara Herrero-Anta (sara@goa.uva.es)

**Abstract.** In the summer of 2019, the Arctic region registered exceptionally high aerosol optical depth (AOD) values over Svalbard, linked to intense biomass burning (BB) and volcanic activity across the Northern Hemisphere. This study presents a comprehensive, multi-instrumental analysis of the aerosol conditions in and around Ny-Ålesund (Spitsbergen, Norway), combining data from ground-based sun-photometry, in-situ observations, active remote sensing (ground-based and on satellite), and atmospheric dispersion modelling (FLEXPART). Despite high AOD was observed during all the period, three different aerosol events are identified in the atmospheric column (6–10 July, 25–28 July, and 6–17 August). In contrast, in-situ surface stations only recorded significant aerosol load during 5–9 July, 30 August, and 12 September, suggesting that most of the aerosol particles remained above the boundary layer. Lidar and photometric observations revealed the presence of spherical, weakly absorbing Accumulation-mode particles (with effective radii between 0.1 and 0.2 $\mu$m) in both the troposphere and stratosphere, with persistent layers extending above 10 km. Simulations carried out with FLEXPART correlate well with the measurements, attributing the observed aerosol events to multiple sources, including Siberian and North American wildfires, the Raikoke (Russia) volcanic eruption, and anthropogenic pollution. While the simulations show a contribution from volcanic aerosols, the contribution from biomass-burning aerosols in the upper troposphere and lower stratosphere were likely more significant under the atmospheric conditions of summer 2019. Overall, the aerosol radiative impact during this long-lasting event was substantial, with a mean reduction in direct solar radiation of approximately -74 $W/m^2$ during July and August. This work shows how the use of dispersion modelling together with multiple observation sources allows to achieve a more complete description of the atmospheric aerosol events and contributes to a better understanding of the overall picture.

# 1 Introduction

The Polar regions constitute sensitive habitats with a vulnerable climate; the particular characteristics of these areas determine
the occurrence of feedback mechanisms that cause an accelerated rate of warming, known as polar amplification (Serreze and
Barry, 2011; Wendisch et al., 2023). The Arctic has warmed nearly four times faster than the rest of the globe since 1979
(Rantanen et al., 2022). This change in climate has an impact in the sources as well as the sinks of aerosols (Willis et al.,
2018a); therefore it is necessary to characterize and understand well the role of natural and anthropogenic pollutants in this
region (Schmale et al., 2021).

In the Arctic, during summer, the aerosol particles typically show smaller sizes and a more local origin (Tunved et al., 2013a;
Schmale et al., 2022; Graßl et al., 2024). However, biomass burning (BB) aerosol from forest fires can also be transported from
lower latitudes (Law and Stohl, 2007; Zielinski et al., 2020). In addition, volcanic eruptions can also sporadically increase the
aerosol concentration in the high Arctic both at ground (Wittmann et al., 2017) and in elevated atmospheric layers (Cheremisin
et al., 2019; Kloss et al., 2021). Long range transport occurs more sporadically and is often confined to specific events. Aerosol
events can occur in the Arctic at any season (e.g. Warneke et al., 2010), but summer events are specially important for the
surface energy budget, since they are characterized by higher solar radiation levels. Aerosol particles also have indirect effects
due to their capacity to act as cloud or ice condensation nuclei, thus affecting clouds properties and formation, and the hydro-
logical cycle, among others (see Lohmann and Feichter, 2005). Lisok et al. (2018) and Calì Quaglia et al. (2022) derived large
negative radiative forcing of BB aerosol for the Arctic during summer events.

In recent years, a large effort has been conducted to monitoring aerosol particles over the Arctic through the deployment of
instrumentation based on different measurement principles, as well as satellite missions. In particular, an important expedition
took place from September 2019 to October 2020: MOSAiC (Multidisciplinary drifting Observatory for the Study of Arctic
Climate; Shupe et al., 2022), the largest Arctic field campaign ever conducted, which, among other data, provided an annual
cycle of aerosol properties over the central Arctic (Ansmann et al., 2023).

To consolidate all available information to date, in and around the Arctic region of Svalbard, the ReHearsol project (Re-
evaluation and Homogenization of Aerosol Optical Depth Observations in Svalbard; Hansen et al., 2022), collected, integrated
and analyzed observations of climate-relevant aerosol parameters like aerosol optical depth (AOD), Ångström exponent (AE)
and black carbon (BC), among others. The outcome of this project was an integrated dataset for the years 2002-2020, which
was analyzed in detail by Hansen et al. (2022) and Hansen et al. (2023). In addition, Xian et al. (2022b) and Xian et al. (2022a)
combined AOD measurements from multi-agency aerosol reanalyses, remote-sensing retrievals, and ground observations avail-
able in the Arctic to analyze the climatology, trends and statistics of extreme events during the 2003-2019 period. This analysis
showed an increase of the maximum AOD values in 2010–2019 compared to 2003–2009, related to stronger wildfire events
in the recent years. In particular, it can be seen in the ReHeaersol dataset that the summer of 2019 clearly stands out, showing
much higher AOD levels than any other period.

The summer of 2019 was a particularly intense fire season in the Northern Hemisphere, mainly due to large wildfires in
North America and Siberia (Fazel-Rastgar and Sivakumar, 2022; Kharuk et al., 2022; CAMS, 2020; Johnson et al., 2021;

Voronova et al., 2020). Antokhina et al. (2023) analyzed the large-scale features of atmospheric circulation to investigate the causes of the natural disasters happening in summer 2019 in Siberia. They found that a severe anticyclonic blocking in Siberia in summer 2019 led to pronounced forest fires in the northern part of Siberia and flooding in the eastern part. This high pressure system might have transported smoke aerosol first northwards into the Arctic and then eastwards towards the American sector.

In addition, the Raikoke volcanic eruption (Kuril Islands, Russia), in June 2019 (Kloss et al., 2021; Vaughan et al., 2021; Cameron et al., 2021; Gorkavyi et al., 2021), increased the background aerosol in the upper troposphere - lower stratosphere (UTLS). For this reason, by the start of MOSAiC campaign in autumn 2019, Ohneiser et al. (2021) expected to find a residual volcanic layer with an AOD of 0.005-0.010 at 500 nm. However, they found a persistent layer of 10 km depth in the UTLS with an AOD of 0.1 and a clear wildfire smoke signature. Their analysis indicated that the conditions during the strong wildfires in Siberia at the end of July and in August were favorable for the self-lofting of the smoke, which was afterwards transported eastward until it reached Europe. Also lidar observations in Leipzig (Germany) corroborated that the smoke of the Siberian wildfires reached the stratosphere. These findings led Pulimeno et al. (2024) to analyze in-situ surface measurements in Ny-Ålesund (Svalbard) during summer and autumn 2019, but this surface analysis did not exhibit any notable peak events. In addition, the chemical analysis revealed only a 2 % contribution of biomass burning to the total concentration averaged during the whole investigated period. Due to the variability of the residence times, removal processes and transport of aerosols in the Arctic, the aerosol contribution in the boundary layer and in the free troposphere is very different (Willis et al., 2018b; Yao et al., 2023). Therefore, surface observations are representative for boundary layer conditions but may not be sufficient to characterize the entire atmospheric column.

With this framework, the main goal of this paper is to identify and characterize the aerosol events that led to the exceptional situation observed in Ny-Ålesund during the summer and autumn of 2019 and to evaluate the radiative impact observed at Ny-Ålesund during this period. For this, all available aerosol-related information for the Arctic region of Svalbard during the summer of 2019 has been compiled in order to conduct a multi-instrumental analysis of the surface and profile aerosol data for the summer of 2019.

This paper is organized as follows: Section 2 provides a brief description of the measurement sites, instrumentation and data used. Section 3 presents the main results: an overview of the temporal evolution of the measurements for different instruments; identification of the periods with aerosol events in the column and surface; analysis of the aerosol properties during these events; determination of the aerosol source using the dispersion model; quantification of the radiative impact. The main conclusions and outlooks are summarized in Section 4.

## 2 Sites, instrumentation and data

### 2.1 Sites

Svalbard is an archipelago in the European Arctic Ocean, right to the east of Greenland, from which it is separated by the Fram strait. The northernmost population in Svalbard is Ny-Ålesund, situated on the west coast of the biggest island of the archipelago, Spitsbergen (see Figure 1). Ny-Ålesund is located in the shore of Kongsfjord, which ends in easterly direction at a

large ice field and is surrounded by glaciers and mountains of about 700 m height. Around Ny-Ålesund, several measurement sites have been established. Within the village, two major sites are located: the joint German-French Arctic station AWIPEV (Alfred Wegener Institute for Polar and Marine Research and the Polar Institute Paul-Émile Victor) and the Ny-Ålesund Research Station Sverdrup, from the Norwegian Polar Institute (NPI). The Zeppelin Observatory (ZEP) is located on top of the Zeppelin Mountain, about 2 km south to the village. ZEP is run by the NPI and the Norwegian Institute for Air Research

(NILU). The Gruvebadet Atmospheric Laboratory (GAL), where the Italian CNR (Consiglio Nazionale delle Ricerche) operates its instrumentation, is located halfway between ZEP and Ny-Ålesund. In the southern part of Spitsbergen, at a fjord with which it shares name, the Institute of Geophysics Polish Academy of Sciences operates the Polish Polar Station, Hornsund. The location of these stations can be seen in Figure 1.

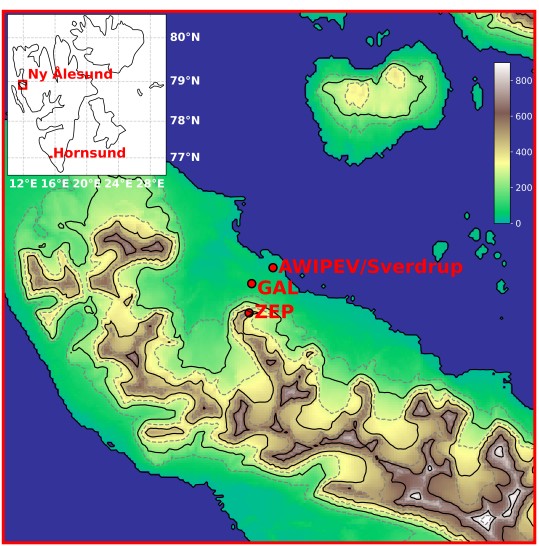

**Figure 1.** Topographic map of the vicinity of Ny-Ålesund and its location on Svalbard. The main stations used for the study have been located in the map: Zeppelin Observatory (ZEP), Gruvebadet Atmospheric Laboratory (GAL), AWIPEV and Sverdrup. The colors indicate the altitude; blue indicates water surface. Map created using the dataset by Moholdt et al. (2019).

In addition to the ground stations, data recorded in summer 2019 onboard the research vessel (R/V) OCEANIA from the

Polish Academy of Sciences has been used. The R/V OCEANIA conducts regular measurement campaigns in the Arctic, and it was travelling through the Fram strait during the period of study. The aerosol measurements from these observatories for the 2002-2020 period from Hansen et al. (2022) built the basis data for this study. This data has been complemented with additional measurements recorded in the same observatories and information from CALIPSO satellite (Cloud-Aerosol Lidar and Infrared Pathfinder Satellite Observation). A summary of the sites, instruments and data products used in the study can be

seen in Table 1. A brief description of the instrumentation, data products and the FLEXPART model setup used for the analysis is given in the following section.

**Table 1.** Summary of sites, instrumentation and data products. Data products include aerosol optical depth (AOD), Backscatter profiles, direct normal irradiance (DIR), diffuse solar irradiance (DIF), aerosol absorption and scattering coefficients, aerosol particle number size distribution and aerosol extinction profiles.

| Site | Coordinates | Instruments | Product |
|---|---|---|---|
| AWIPEV | 78.92°N, 11.92°E | SP1A | AOD 500 nm |
| | 7 m a.s.l. | KARL | Backscatter profiles at 355, 532 and 1064 nm, polarization at 532 nm |
| | | CHP1 Pyrheliometer | DNI |
| | | CMP22 Pyranometer | DIF |
| Sverdrup | 78.92°N, 11.93°E | PFR | AOD 500 nm |
| | 10 m a.s.l. | | |
| GAL | 78.92°N, 11.89°E | PSAP | Aerosol absorption coefficient |
| | 20 m a.s.l. | Nephelometer | Aerosol scattering coefficient |
| | | SMPS, APS | Aerosol particle number size distribution |
| ZEP | 78.91°N, 11.89°E | SP1A | AOD 500 nm |
| | 474 m a.s.l. | MAAP | Aerosol absorption coefficient |
| | | Nephelometer | Aerosol scattering coefficient |
| | | DMPS | Aerosol particle number size distribution |
| Hornsund | 77.00°N, 15.54°E | CE318-T | AOD 500 nm, AE 440-870, volume size distribution and single scattering albedo at 440, 675, 870 and 1020 nm |
| | 12.5 m a.s.l. | | |
| R/V OCEANIA | 67°N-85°N, 20°W-20°E | Microtops II | AOD at 500 nm |
| | 0 m a.s.l. | | |
| CALIPSO | < 50 km from Ny-Ålesund | CALIOP | Aerosol extinction profiles at 532 nm, tropospheric and stratospheric AOD at 532 nm, tropopause altitude |

## 2.2 Instrumentation and data

### 2.2.1 Sun-photometers

Sun-photometers measure direct sun spectral irradiance, which is used to derive the AOD at different wavelengths. AOD is an indicator of the amount of aerosol in the atmospheric column. In addition, the Ångström exponent (AE) can be calculated from the spectral dependence of the AOD for the range of wavelengths available. AE is related to the aerosol particle size (Kokhanovsky, 2008). A mean value of AE equal to 1.3 is observed for the average continental aerosol (Ångström, 1929), while values close to 0 indicate coarse particles.

Different models of sun-photometers are distributed around Svalbard. A precision filter radiometer (PFR; Kazadzis et al., 2018; Kouremeti et al., 2022) and a sun photometer 1 automatic (SP1A; Herber et al., 2002; Mazzola et al., 2012; Graßl and Ritter, 2019) are located at the Sverdrup and the Zeppelin Observatory respectively. In Hornsund, there is a CE318-T (Cimel Electronique) belonging to the Aerosol Robotic Network (AERONET; Holben et al., 1998). CE318-T instruments can also measure sky radiances at various wavelengths, which are then used together with simultaneous AOD measurements to retrieve microphysical and optical aerosol properties using an inversion algorithm (Sinyuk et al., 2020). Finally, a Microtops II manual photometer onboard the R/V OCEANIA was used to complement the fixed location measurements with AODs from different locations around the Svalbard Archipelago. This instrument is part of the maritime aerosol network from AERONET (Smirnov et al., 2009), and is used to provide ship-borne AOD measurements. The Microrotops II sun photometer technical parameters as well as calibration techniques can be consulted in Morys et al. (2001) and Markowicz et al. (2012). During the 2019 study, each measurement slot lasted for about 1 minute. In order to reduce potential human errors (e.g. sun pointing error), 5 scan slots with each measurement and the scan with the lowest standard deviation was further regarded for the processing analyses. For this study only data recorded when the vessel was 50 km distance or less from Ny-Ålesund have been used.

The sun-photometer dataset used in this study from Hansen et al. (2022) includes the AOD at several wavelengths (depending on the instrument) and the AE derived in the range 380-870 nm. The comparability of the AOD from the different sun-photometers was already assessed within the mentioned projects. For this analysis, hourly and instantaneous values of AOD at 500 nm and AE in the 380 and 870 nm range have been used. In addition, particle volume size distribution (PVSD) and single scattering albedo (SSA) have been directly obtained from the AERONET website (https://aeronet.gsfc.nasa.gov/, last accessed December 17, 2025). All AERONET data used correspond to level 1.5 products (version 3). These retrieved properties have been filtered by the residual obtained by the inversion algorithm; inversions with residuals bigger than 10 % have been rejected, which ensures the quality of the retrievals. For the period of study the PVSD and SSA are only available at Hornsund.

### 2.2.2 In-situ observations

In-situ aerosol optical and microphysical properties are measured continuously at ZEP and GAL. At these two sites the aerosol absorption coefficient ($\beta_{abs}$) and the aerosol scattering coefficient ($\beta_{sca}$) both at 530 nm, and the particle number size distribu-

tion (PNSD) between 10 – 700 nm are recorded. In addition, the single scattering albedo at the same wavelength (SSA$_{530}$) can be calculated as the ratio between the scattering coefficient and the scattering coefficient plus the absorption coefficient.

At ZEP, $\beta_{abs}$ was originally measured at 637 nm using a multi-angle absorption photometer (MAAP, Thermo Fisher Scientific Inc., Germany, model 5012), and after converted to 530 nm (assuming an AE of 1). $\beta_{sca}$ was recorded using a nephelometer (Ecotech Pty Ltd., Australia, model Aurora 3000), measuring at 450 nm, 550 nm and 700 nm, and converted to 530 nm using the scattering Ångström exponent (SAE). The PNSD was measured using a twin differential mobility particle sizer (DMPS) system, consisting of two custom built Hauke-type differential mobility analysers (DMAs) each connected to a condensational particle counter (CPCs a TSI 3010 CPC and TSI 3772) used to count the size discriminated aerosol particles. All samplings were conducted using a whole-air inlet in accordance with sampling procedures laid out by the World Meteorological Organization / Global Atmosphere Watch (WMO/GAW) guidelines.

At GAL, $\beta_{abs}$ was recorded with a Particle Soot Absorption Photometer (PSAP, Radiance Research) at three wavelengths (467 nm, 530 nm, 660 nm), while $\beta_{sca}$ was measured at 530 nm with a Nephelometer (Radiance Research). Raw data were corrected, averaged over 1-hour, and converted to standard temperature and pressure conditions as described by Gilardoni et al. (2023). The aerosol PNSD was continuously measured at GAL using a combination of a Scanning Mobility Particle Sizer (SMPS) model TSI 3034 and an Aerodynamic Particle Sizer (APS) model TSI 3321, both from TSI (see Rinaldi et al., 2021, and references within).

### 2.2.3 KARL lidar

The lidar measurements were performed by the Koldewey Aerosol Raman Lidar (KARL) at AWIPEV. This system has a Spectra 290/50 Nd:YAG laser, which emits a laser pulse at 355 nm, 532 nm and 1064 nm with 50 Hz and 200 mJ per colour vertically into the atmosphere. A telescope with 70 cm diameter collects backscattered photons with a field of view of about 2 mrad. Data recording is done via Hamamatsu photomultipliers and Licel transient recorders with a raw resolution of 7.5 m and 82 s. Both, analogue and photo counting signals, are recorded. Full overlap is reached at about 700 m altitude. In addition, KARL receiver can separate the backscattered signal into parallel and perpendicular components at 532 nm, which allows the calculation of the volume aerosol depolarization ($\delta^{aer}$) as the ratio between the perpendicular and the parallel signal. This magnitude is useful to distinguish between spherical and non-spherical particles. A more detailed description of KARL can be consulted in Hoffmann (2011).

The aerosol backscatter profiles at the three available wavelengths have been calculated with 60 m and 600 s resolution according to Klett (Speidel and Vogelmann, 2023) with clear sky approximation (aerosol backscatter small against molecular backscatter at altitudes > 22 km) and the choice of a prescribed lidar ratio (Ritter and Münkel, 2021) of 70, 45 and 45 sr for the wavelengths of 355, 532 and 1064 nm, respectively. The lidar ratios for 355 and 532 nm have been verified by backscatter values in the clear troposphere. An uncertainty of ±10 sr for the lidar ratio has been estimated, giving rise to about 10% uncertainty in the derived aerosol backscatter. For the 1064 nm wavelength the uncertainty is dominated by the assumed backscatter > 22 km as a boundary condition, such that also 10% uncertainty at this wavelength is realistic. Data points in time and altitude which were covered by clouds have been removed to not bias aerosol properties.

The backscatter profiles have been used to calculate the colour ratio (CR), for each $((\lambda_1, \lambda_2))$ combination of the three laser wavelengths as Equation 1:

$$CR(\lambda_1, \lambda_2) = \frac{\beta^{aer}(\lambda_1)}{\beta^{aer}(\lambda_2)} \qquad \text{with } \lambda_1 < \lambda_2 \tag{1}$$

To adjust for measurement uncertainties, an uncertainty of $\pm 5\ \%$ of the CR has been considered. The CR can also be theoretically estimated using the Mie calculator from Library for Radiative Transfer – libRadtran (Mayer and Kylling, 2005; Emde et al., 2016). For this calculus we have assumed a log-normal particle size distribution with a width of $\sigma = 1.1$ and a refractive index of $m = 1.5 + 0.05i$; which is a typical refractive index for Arctic aerosol (Böckmann et al., 2024). According to the previous work from Böckmann et al. (2024) the choice of the refractive index will probably not critically affect the solution. The best fit between the theoretical and observed colour ratios for each combination of the available wavelengths 355, 532 and 1064 nm has been used to estimate the height-dependent effective radius of the aerosol. This way, vertical profiles of the aerosol effective radius have also been obtained.

### 2.2.4 Surface radiation

The set-up of surface radiation observations established at AWIPEV includes direct normal irradiance (DNI) by a CHP1 pyrheliometer, diffuse (DIF), global and reflected shortwave horizontal solar irradiance by CMP22 pyranometers, as well as up and downward longwave radiation by CGR4 and Eppley PIR pyrgeometers. All instruments follow the World Radiometric Reference (WRR) standards for solar radiation data and the World Infrared Standard group (WISG) guidelines for longwave radiation. Additionally, the data contributes to the Global Climate Observing System (GCOS) Reference Upper-Air Network (GRUAN). The surface radiation measurement setup, data retrieval, and quality control follow Baseline Surface Radiation Network (BSRN) standards. More information is given by König-Langlo et al. (2013) and Maturilli et al. (2015), among others. All measurements were performed automatically with a 1-min resolution and can be found in Maturilli (2020) (https://doi.org/10.1594/PANGAEA.914927). These data are available since 2006.

The hourly anomaly ($\Delta_{DNI}$) for the hourly DNI values has been calculated with respect to a reference value ($DNI_{ref}$), as shown in Equation 2:

$$\Delta_{DNI} = DNI - DNI_{ref} \tag{2}$$

The $DNI_{ref}$ values have been calculated for each month and each hour as the mean of all DNI hourly values recorded at that specific hour during that month from the 2006 to 2020 reference period. As a result, 24 reference values are obtained for each month. This approach is used because, due to the high frequency of cloudy conditions in the region, the amount of available data to construct a reliable reference is limited.

In addition, the diffuse ratio (DifR; see Long et al., 1995) has been calculated following Equation 3:

$$\text{DifR} = \frac{\text{DIF}}{\text{DIF} + \text{DNI} \cdot \cos(\text{SZA})} \tag{3}$$

This ratio can take values from zero to unity, with small values related to cloud-free cases. Only data showing an hourly DifR < 0.3 have been used, trying to reject the influence of clouds on the results.

### 2.2.5 CALIOP

CALIOP (Cloud-Aerosol LIdar with Orthogonal Polarization) is the main instrument onboard CALIPSO satellite. CALIPSO was launched on April 2006 (Winker et al., 2009) and after 17 years of operation, it ended its mission on 1 August 2023.

CALIOP consists of a two-wavelength lidar that uses a Nd:YAG laser to emit pulses at 1064 and 532 nm simultaneously. The laser pulse repetition frequency is 20.16 Hz, which allows a horizontal resolution of 335 m at the ground. The outgoing 532 nm pulses are linearly polarized. A polarization beam splitter separates the components of the 532 nm backscatter signal polarized parallel and perpendicular to the plane of the outgoing beam, which are collected by two detectors. The instrument operates continuously, providing observations during both day and night.

In this work it has been utilized the level 2 (L2) profile data products (LID_L2_05kmAPro-Standard-V4-51) for aerosol extinction at 532 nm, which is given at a spatial resolution of 60 m vertically and horizontal averaging length of 5 km along the satellite track. The L2 layer data product (LID_L2_05kmALay-Standard-V4-51) includes the tropospheric and stratospheric AOD at 532 nm, as well as the L2 vertical feature mask (LID_L2_VFM-Standard-V4-51) for aerosol and cloud classifications. The classification (read out from the vertical feature masks of the profile) and the lidar ratios, which are used by the CALIOP retrieval teams for conversion of the measured backscatter into the estimated extinction have been included in Table S1 in the supplement.

CALIPSO data have been downloaded from the ICARE Data and Services Center (http://www.icare.univ-lille1.fr/, last access: 23 August 2023). For this analysis, data within 50 km distance from Ny-Ålesund have been extracted, and the closest extinction profile has been utilized (see Figure S1 in the supplement).

### 2.2.6 Atmospheric dispersion modelling/FLEXPART

To investigate the possible origin of BC (black carbon) aerosol air-masses reaching the Arctic, the Lagrangian particle dispersion model FLEXPART version 10.4 has been used (Pisso et al., 2019). The model has been driven by hourly reanalysis meteorological fields (ERA5) from the European Centre for Medium-Range Weather Forecasts (ECMWF) with 137 vertical levels and a horizontal resolution of $0.5° \times 0.5°$ (Hersbach et al., 2020). Computational particles have been released in FLEXPART at various heights (0 - 100, 100 - 500, 500 - 4000, 4000 - 6000, 6000 - 10000 m) from the receptor (considered the Zeppelin Observatory) and tracked backward in time in FLEXPART's "retroplume" mode. Simulations have been run 30 days back in time, that is a sufficient time to include most BC sources arriving at the station, given a typical BC lifetime of 1 week (Bond et al., 2013).

Gravitational sedimentation for spherical particles is included and differs between trajectory models due to its ability to simulate dry and wet deposition of aerosols (Grythe et al., 2017), turbulence (Cassiani et al., 2015), unresolved mesoscale motions (Stohl et al., 2005), and convection (Forster et al., 2007). In the simulations of this work it has been assumed that BC has a density of $1500\,\text{kg}\,\text{m}^{-3}$ and follows a logarithmic size distribution with an aerodynamic mean diameter of $0.25\,\mu\text{m}$ and a logarithmic standard deviation of 0.3 (Long et al., 2013).

Anthropogenic emissions have been adopted from the latest version (v6b) of the ECLIPSE (Evaluating the CLimate and Air Quality ImPacts of ShortlivEd Pollutants) emission inventory, an upgraded version of the previous version (Klimont et al., 2017). This inventory includes industrial combustion emissions, emissions from the energy production sector, residential and commercial emissions, emissions from waste treatment and disposal sector, transportation, shipping activities and gas flaring emissions. Biomass burning emissions have been adopted from the Copernicus Global Fire Assimilated System (CAMS GFAS) (Kaiser et al., 2012) because this product provides an estimation of the injection altitude of the fire emissions. Smoke particles consist mainly of organic carbon (OC) and a small amount of BC (Ohneiser et al., 2023). These components can absorb radiation, warming the surrounding air and inducing upward motion that lifts the aerosol (Johnson and Haywood, 2023); i.e., the so-called self-lofting mechanism. GFAS uses two different models to calculate the injection height, based on satellite observed FRP and ECMWF forecasts of key atmospheric parameters (Rémy et al., 2017). Radiative self-lofting in global models, such as FLEXPART, is not considered yet, but the scientific basis now exists with the ECMWF radiation scheme (ECRAD) that computes shortwave heating rates of an imposed smoke layer (Ohneiser et al., 2023). However, online implementation of this module in global models might be demanding, due to the need of remote sensing data as input parameters (e.g., CALIOP aerosol observations, MODIS aerosol optical depth retrievals etc.). A more detailed discussion of the potential mechanisms responsible for self-lofting is provided in the FLEXPART results section (see Section 3.2.1).

FLEXPART simulations have been performed every 3 hours during the studied period. The FLEXPART output consists of a footprint emission sensitivity, which results in a modelled concentration at the receptor (Zeppelin Observatory), when coupled with gridded emissions from an emission inventory (like the ones described before). The emission sensitivity expresses the probability of any release occurring in each grid-cell to reach the receptor. The modelled concentrations can be displayed as a function of the time elapsed since the emission has occurred (i.e., "age"), which can be shown as an "age spectrum". Detailed source contribution maps have been calculated, showing which regions contributed to the simulated concentration.

The source contributions to receptor BC have been derived by combining each gridded emission sector (e.g. gas flaring, transportation, waste management, etc) from an emission inventory with the footprint emission sensitivity. Calculations for anthropogenic sources and open biomass burning have been performed separately. This has enabled the identification of the exact origin of BC and allowed for quantification of its source contribution.

## 3 Results

### 3.1 General overview

The data obtained from the various instruments described in the previous section have been analyzed to characterize the aerosol reaching Svalbard in summer 2019. First, an analysis of the columnar aerosol load has been conducted using sun-photometry. Next, data from in-situ observatories have been used to analyze the aerosols reaching the surface. Finally, active remote sensing instrumentation has been used to identify the altitude at which the aerosol intrusion is occurring.

#### 3.1.1 Columnar aerosol

Figure 2 shows the hourly AOD values at 500 nm ($AOD_{500}$) recorded in and around Svalbard from June to September 2019. The mean daily $AOD_{500}$ values calculated considering all the available period (2002-2020) are included as reference (see dashed line in Figure 2). It can be seen that most of the summer of 2019 (considered from June to September for this study, if nothing else is specified) presented consistently higher values of AOD with respect past and following years (2002 to 2020 period). $AOD_{500}$ in 2019 was similar to the reference values during June, but since 6 July, when an unprecedented increase in

AOD occurred, it remained high for the rest of the summer. The median $AOD_{500}$ from July to September for the 2002-2020 reference period is 0.048, but this value is 0.162 when it is calculated only with 2019 data; it means that $AOD_{500}$ in this period was more than three times higher than during the reference. Regarding the AE, which is related to the size of the aerosol, it did not show a significant difference with respect to the reference period as can be observed in Figure S2 in the supplement. The days on which the $AOD_{500}$ values were higher than the summer average plus three times the standard deviation have

been identified as days when an aerosol intrusion was occurring in Svalbard. This way, three different aerosol events have been identified during the summer of 2019 in the following periods: 6 to 10 July, 25 to 28 July and 6 to 17 August, which have been named C1, C2 and C3 respectively because they are observed in the atmospheric column (C). During these three events the mean $AOD_{500}$ values were 0.28, 0.19 and 0.23 respectively. Moreover, the periods between C1 and C2, and between C2 and C3 presented mean $AOD_{500}$ values of 0.15 and 0.17 respectively; this mean value was 0.15 from the end of event C3 (17

August) to the end of August. Finally, during September the mean $AOD_{500}$ was 0.11.

The columnar PVSD and SSA values retrieved for the analyzed period are shown in Figure 3, highlightening those corresponding to the identified events. Table 2 summarizes the mean AOD and mean parameters of the PVSD, observed during the periods identified with aerosol event, and also in the adjacent periods during summer 2019. Figure 3a-d shows a high aerosol concentration in fine mode (Accumulation mode) during July and August; this also happens for September, when no aerosol

events have been identified. Aitken mode and coarse mode particles are almost absent; the highest variability in PVSD during this summer is observed in the Accumulation mode, when the maximum volume concentrations of the PVSD oscillate from 0.01 to 0.04 $\mu m^3 \mu m^{-2}$ in all the period.

In July, the mean effective radius of the fine mode ($< R_{eff}F >$) was 0.145 $\mu m$ the days with no aerosol event, and 0.156 $\mu m$ and 0.155 $\mu m$ in events C1 and C2, respectively. However, while C1 showed a slim fine mode distribution with one maximum,

**Table 2.** Mean values of the aerosol optical depth (AOD) and the parameters of the PVSD: effective radius ($R_{eff}$), standard deviation ($\sigma$), and volume concentrations (VC), for fine (F) and coarse (C) modes, and the total volume concentration (VCT), during the different periods identified with aerosol event, and adjacent periods during summer 2019 (June to September). Products extracted from AERONET.

| | AOD | $R_{eff}$F [$\mu$m] | $\sigma$F [$\mu$m] | VCF [$\mu$m$^3\mu$m$^{-2}$] | $R_{eff}$C [$\mu$m] | $\sigma$C [$\mu$m] | VCC [$\mu$m$^3\mu$m$^{-2}$] | VCT [$\mu$m$^3\mu$m$^{-2}$] |
|---|---|---|---|---|---|---|---|---|
| **June** | 0.05 | 0.149 | 0.448 | 0.005 | 1.486 | 0.707 | 0.006 | 0.012 |
| **C1 (6-10 July)** | 0.28 | 0.156 | 0.476 | 0.024 | 1.728 | 0.634 | 0.004 | 0.029 |
| **11-24 July** | 0.15 | 0.145 | 0.537 | 0.037 | 2.328 | 0.663 | 0.002 | 0.039 |
| **C2 (25-28 July)** | 0.19 | 0.155 | 0.578 | 0.041 | 2.401 | 0.625 | 0.004 | 0.045 |
| **29 July - 5 August** | 0.17 | - | - | - | - | - | - | - |
| **C3 (6-17 August)** | 0.23 | 0.182 | 0.585 | 0.035 | 2.166 | 0.654 | 0.005 | 0.04 |
| **18-31 August** | 0.16 | 0.156 | 0.617 | 0.04 | 2.354 | 0.707 | 0.003 | 0.042 |
| **September** | 0.11 | 0.142 | 0.618 | 0.033 | 2.609 | 0.576 | 0.005 | 0.037 |

C2 generally showed a main maximum at radius about 0.1 $\mu$m and a second maximum at about 0.3 $\mu$m. This might indicate the presence of particles of different origin during C2.

In August, the fine mode distribution was wider and with a gaussian shape, presenting in general lower concentrations at each radius. Event C3 presented the highest $< R_{eff}F >$ (0.182 $\mu$m) of the three identified events. This might either point to different aerosol source or be an aging effect, such that the mean time between the aerosol emission and arrival over Spitsbergen might had been longer in August compared to July. These larger radii made AE values lower in event C3 (see Figure S2) than in the other events even when all were dominated by fine particles (González et al., 2020). Nevertheless, days not identified as aerosol intrusion showed a distribution similar to the ones observed during event C2, with a mean $< R_{eff}F >$ of 0.156 $\mu$m.

In September only two inversions are available (on the 5[th]). These presented a $< R_{eff}F >$ of 0.142 $\mu$m and, again, a similar shape, but with a lower concentration at each radius than the ones observed in event C2 (July). However, the coarse mode seems slightly enhanced and these particles, larger than 2 $\mu$m, should sediment down quickly. As similar observations are made throughout July, August and September, this might indicate a steady supply of new aerosol (because of the life-time of this coarse particles). Alternatively, due to the low concentration of the coarse mode, it could just indicate the presence of a steady stratospheric aerosol starting from event C2, which was combined with new aerosol with the arrival of new aerosol events. Therefore it can be hypothesized that we are facing a combination of different aerosol sources.

The SSA, shown in Figure 3e-h, presents values close to 1 for all wavelengths, indicating a low aerosol absorption even during the aerosol events. Only a few background cases showed values lower below 0.96 in the infrared, but these cases do not show up for the identified events, indicating less variability. The high SSA and no spectral dependency during the events could be caused by the presence of BB aged after long transport (Zielinski et al., 2020) or might have been emitted under low combustion efficiencies (Liu et al., 2014). This kind of aerosol agrees well with the predominant fine mode with an effective radius larger than usual (González et al., 2020).

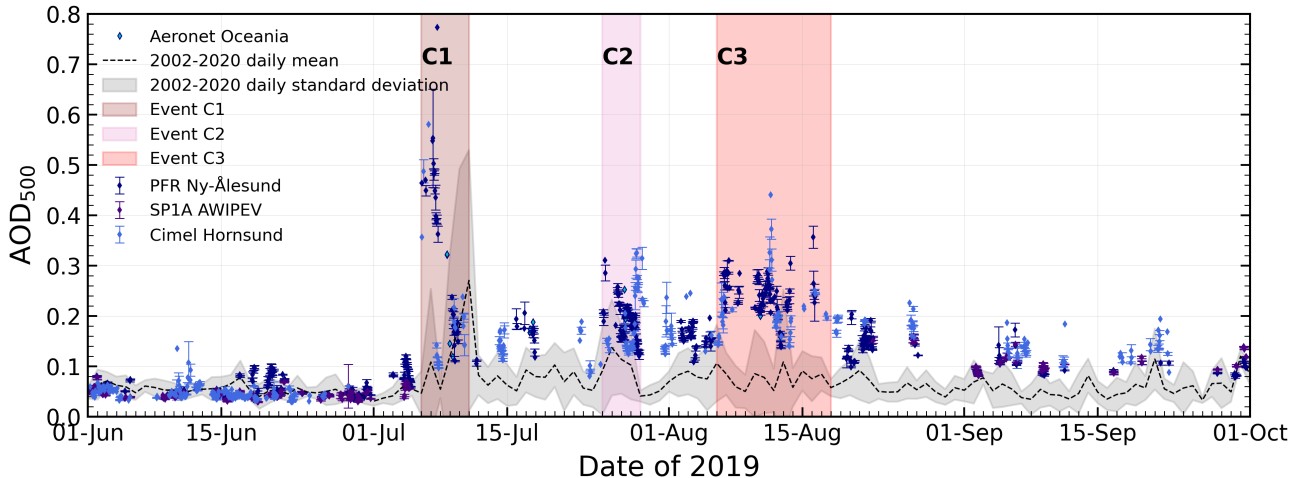

**Figure 2.** Aerosol Optical Depth (AOD) temporal evolution (hourly means with standard deviation) in and around Svalbard in the summer of 2019 (June to September), with data from Ny-Ålesund, Hornsund and R/V OCEANIA (in the Fram Strait). The time periods highlighted in dark red, pink and red shaded areas correspond to the three events identified as aerosol events in the column, respectively: 6-10 July (C1), 25-28 July (C2) and 6-17 August (C3). The scatter points with error bars represent hourly mean values and their associated standard deviation. The black dashed line represents the mean daily values of AOD at 500 nm calculated with all the available measurements on the reference period, i.e. from 2002 to 2020; the grey shaded area represents the corresponding standard deviation. The high reference values observed on 10 and 11 July are due to an extreme event that occurred in 2015.

### 3.1.2 Surface aerosol

Figures 4a and b report, respectively, the time series of the aerosol absorption ($\beta_{\mathrm{abs}}$) and the scattering ($\beta_{\mathrm{sca}}$) coefficients at 530 nm measured in-situ at GAL and ZEP in summer 2019. During the full period, $\beta_{\mathrm{abs}}$ and $\beta_{\mathrm{sca}}$ showed the following mean values: $0.12 \pm 0.16\,\mathrm{Mm}^{-1}$ and $1.83 \pm 2.04\,\mathrm{Mm}^{-1}$ at GAL and $0.10 \pm 0.21\,\mathrm{Mm}^{-1}$ and $2.43 \pm 2.97\,\mathrm{Mm}^{-1}$ at ZEP, respectively. The average $\mathrm{SSA}_{530}$ was $0.92 \pm 0.06$ and $0.96 \pm 0.06$ at GAL and ZEP, respectively. The seasonal averages for the summers of 2018 and 2020 together with the ones from 2019 are included in Table 3. It shows that 2019 did not show significant differences at ground level in these type of measurements.

The $B_{abs530}$ at GAL and ZEP correlate with a Pearson coefficient (R) of 0.71. A lower correlation is observed for $B_{sca530}$, with an R of 0.34, although increasing and decreasing trends have been generally observed at the same time at the two sites. This low R value points to a more local origin for scattering aerosol. While absorption is predominantly associated with long-range transport, the scattering in summer appears to have a strong marine influence. Consequently, when wind directions differ between the two sites, the stations may perceive this local source differently. However, whenever a strong event comes, the scattering increases in both sites. Therefore, it is difficult to use only one station for comparison with climate models.

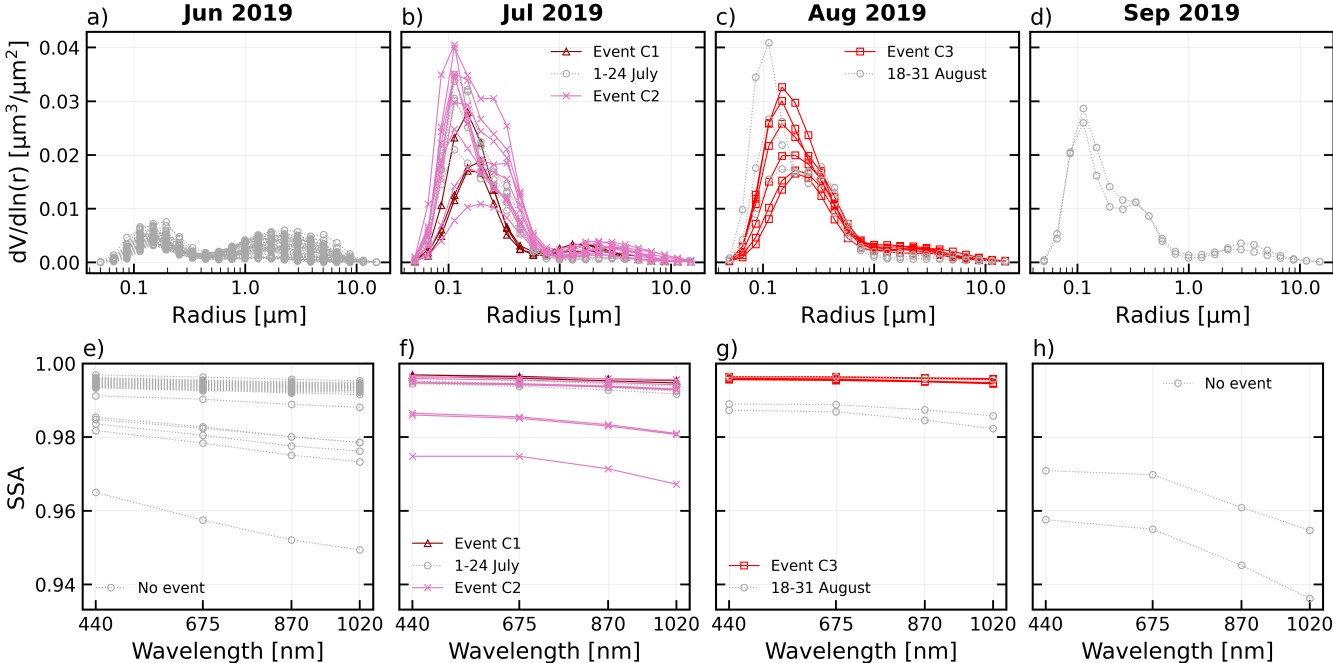

**Figure 3.** a-d) Columnar particle volume size distribution and e-h) Single Scattering Albedo (SSA) AERONET products available from from June to September 2019 at Hornsund. The properties corresponding to the aerosol events identified in Figure 2 have been highlighted with different markers and colours.

**Table 3.** Seasonal averages (and standard deviation in parenthesis) for summers (June to September) 2018, 2019 and 2020 for aerosol absorption ($\beta_{\mathrm{abs}}$) and the scattering ($\beta_{\mathrm{sca}}$) coefficients at 530 nm.

| | GAL | | ZEP | |
|---|---|---|---|---|
| Year | $\beta_{\mathrm{abs}530}$ | $\beta_{\mathrm{sca}530}$ | $\beta_{\mathrm{abs}530}$ | $\beta_{\mathrm{sca}530}$ |
| **2018** | 0.105 (0.143) | 3.696 (2.492) | 0.068 (0.077) | 2.787 (2.479) |
| **2019** | 0.120 (0.163) | 1.833 (2.045) | 0.099 (0.207) | 2.433 (2.966) |
| **2020** | 0.161 (0.245) | 2.845 (2.884) | 0.115 (0.245) | 1.636 (1.230) |

Aerosol events at the surface, associated with relatively high aerosol loading at the ground, have been identified when both $B_{abs530}$ and $B_{sca530}$ exceeded, at least in one station, the average values recorded over three summers (2018-2020, from June to September), plus three times the standard deviation (horizontal lines in Figure 4). This way, three events of high aerosol load on the surface (S) have been identified on 5 to 9 July, 30 August, and 12 September. The first event (5 to 9 July) overlaps in time with event C1, therefore we will refer to it as CS1 instead of C1 from now on. The last two events have been named events S1 and S2 respectively. No data from remote sensing observations were available during these two events due to cloudy conditions.

The increase in absorption coefficient observed during the three surface events points to the impact of air masses affected by combustion emissions. The $SSA_{530}$ observed during events S1 and S2 were $0.89\pm0.04$ and $0.92\pm0.05$, respectively, reaching values as low as 0.81 (event S1) and 0.76 (event S2). For event CS1 the mean $SSA_{530}$ was $0.95\pm0.01$ with lowest instantaneous values equal to 0.92. The slightly lower $SSA_{530}$ values observed during the events S1 and S2 might indicate that the aerosol particles emitted by combustion traveled a shorter distance before reaching Svalbard compared to event CS1, although the differences are within the standard deviations.

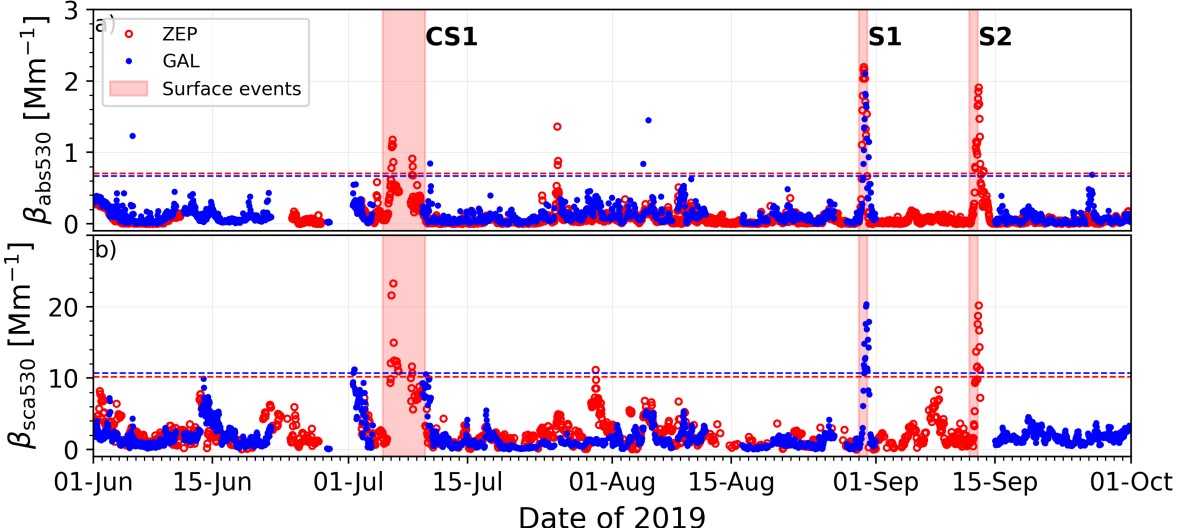

**Figure 4.** Time series of a) aerosol absorption coefficient ($\beta_{abs}$) and b) scattering coefficient ($\beta_{sca}$) at 530 nm at Gruvebadet (GAL), in blue, and Zeppelin (ZEP), in red. The horizontal lines correspond to the average of the logarithm of scattering and absorption coefficients over three summers (2018-2020, from June to September) plus three times the standard deviation (in blue for GAL and in red for ZEP) used to identify aerosol events. The time periods with red shaded areas correspond to the three events identified as high aerosol load in the surface: 5 to 9 July (CS1), 30 August (S1), and 12 September (S2).

Figure 5 reports the averaged particle number size distribution during the surface events and for all available measurements in the summer of 2019. The PNSD values observed at the two sites align with the multi-year average distributions reported by Dall'Osto et al. (2019) for the summer months. Regarding GAL observations, the seasonal average of PNSD exhibits a bimodal pattern, with peaks occurring at approximately 30 nm (Aitken mode) and 140 nm (Accumulation mode). The Aitken mode predominates over the Accumulation mode, primarily due to new particle formation events that increase the number of ultrafine particles (Dall'Osto et al., 2019; Tunved et al., 2013b). Additionally, wet scavenging removes effectively Accumulation mode particles in summer both in the Arctic and during their transport to the Arctic (Dall'Osto et al., 2019; Gilardoni et al., 2019). The surface events were still characterized by a bimodal distribution, but in these cases the particle number of the Accumulation mode was comparable or higher than the Aitken mode population, indicating that the site was reached by more aged aerosol

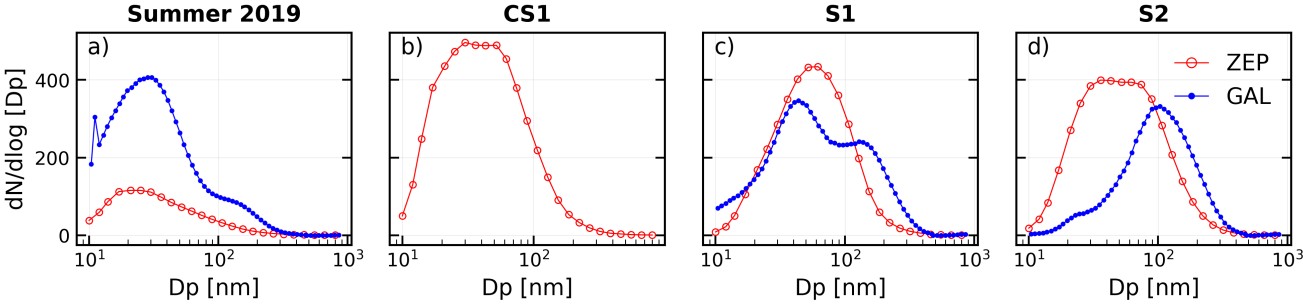

**Figure 5.** Average particle number size distribution (PNSD) at Gruvebadet (GAL), in blue, and Zeppelin (ZEP), in red, during the summer of 2019 (June to September) (a) and events CS1 (b), S1 (c) and S2 (d). For CS1 there are only data from ZEP.

populations. The increase in Accumulation mode particles agrees with the rise in scattering coefficients during the three events compared to the average. The PNSD at ZEP does not show a clear bimodal structure for either of the events. In these cases, the peak of the distribution lies between the Aitken and Accumulation modes, generally showing higher concentrations than GAL during events S1 and S2. Therefore, the surface events were perceived slightly differently at the two sites.

### 3.1.3 Aerosol vertically-resolved properties

A time series with the CALIOP aerosol extinction profiles (at 532 nm) available in summer 2019 is shown in Figure 6. The equivalent Figure S3 in the supplement indicates the classification of the layer aerosol type. The tropospheric and stratospheric AODs retrieved from CALIPSO for each profile are also included in the upper panel. For event CS1 a strong extinction is observed between 3 and 5 km a.g.l. Starting from 15 July a stratospheric aerosol layer at about 12 km a.g.l. is observed, with the corresponding increase in the stratospheric AOD. An aerosol layer between 10 and 15 km a.g.l. is also observed in the following extinction profiles, with variable intensity. During C2 and C3, no clear signature of aerosols is observed in the troposphere, maybe related to the presence of tropospheric clouds, which would mask the aerosol when observing from a satellite.

Unfortunately, KARL measurements are only available for four days in the summer of 2019 due to cloudy conditions and safety regulations of the instrument. The layers observed with KARL were temporally quite constant on each day (See Figure S4 in the supplement), therefore, the daily averaged backscatter profiles have been calculated. These are shown in Figure 7. It is observed and increased backscatter between 10 and 16 km a.g.l. with several layers through August, but in 17 September the backscatter slowly decreases and becomes more homogenous with height. Only one of these days corresponds to a day identified with aerosol event, 11 August (Event C3). During this day, a high backscatter coefficient at 532 nm up to about 0.8 $Mm-1$, with several layers, is observed throughout the entire troposphere, as well as in the stratosphere up to nearly 16 km a.g.l.. Particularly, the layer just around the around 10 km a.g.l. observed with KARL correlates very well in altitude with the increased backscatter profile measured by CALIOP in the same date In CALIOP it is also observed some extinction

around 14 km a.g.l., which correlates with the stratospheric layers observed with KARL. Since the vertical and temporal resolution from both instruments is very different, we do not expect a closer agreement.

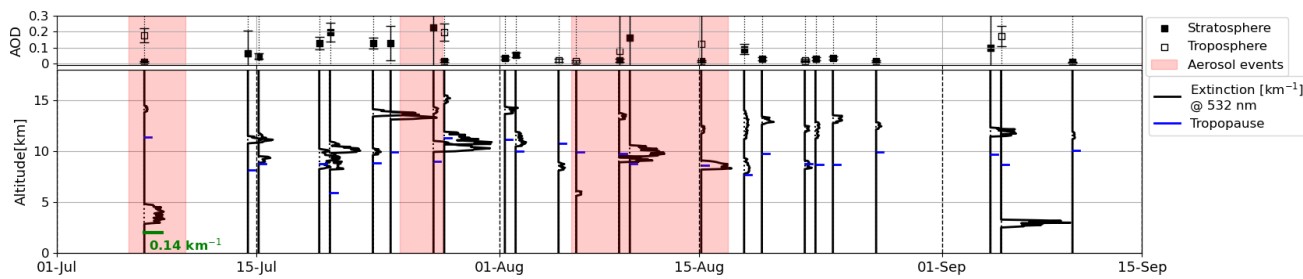

**Figure 6.** Time series of the extinction profiles at 532 nm measured by CALIOP in the summer of 2019. For each time period where aerosol layer was identified, the enhancement of the extinction within the layer is shown; the zero line indicated the date-time of observations. For reference, an x-scale for the extinction profiles has been included in green in the first profile. The blue lines indicate the tropopause. The tropospheric and stratospheric AOD at 532 corresponding to each profile is included in the upper panel; the corresponding uncertainty of the AOD is given by the bars. The red shaded areas indicate the days on which the columnar events were identified (CS1, C2 and C3).

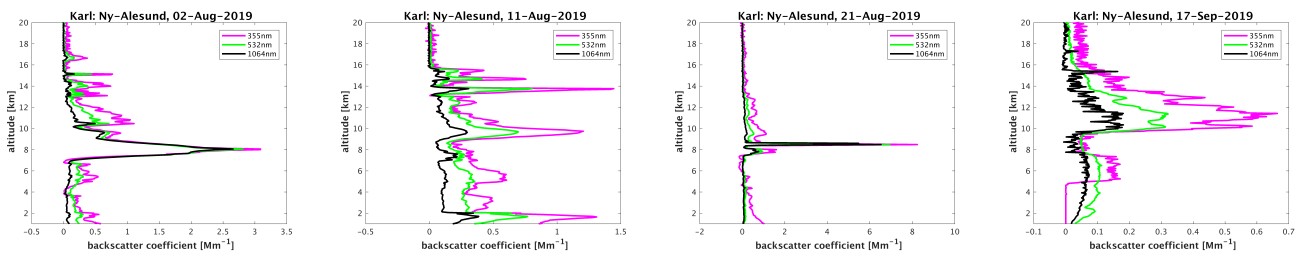

**Figure 7.** Daily averaged backscatter coefficient profiles from KARL on 2, 11 and 21 August and 17 September. The pink, green and black lines correspond, respectively, to the signal at 355, 532 and 1064 nm.

The daily averaged backscatter profiles have been used to estimate the height-dependent effective radius of the aerosol for each day, as described in Section 2.2.3: an a priori refractive index and a one-modal log-norm distribution are considered. The calculated effective radii for the four observation days are shown in Figure 8 with a height resolution of 60 m. Due to the incomplete lidar overlap, the plot starts at 1 km altitude. This figure shows that for all days and the entire troposphere the background aerosol had radii values at around 0.1 $\mu$m, while the pronounced layers reveal coarser aerosols with radii from

0.5 $\mu$m to 0.8 $\mu$m. Furthermore, it can be seen that the effective radius for August was very similar, while on 17 September, between 7 and 9 km a.g.l., the effective radius was about 0.2 $\mu$m smaller. This suggests that the largest particles were removed from this layer, possibly due to sedimentation. Depolarization measurements at 532 nm ($\delta^{aer} < 3$ %) were lower than 3 % for the four available days. These low values are typical for a spherical shape. Hence, we conclude that the large particles in the troposphere were likely droplets. Hygroscopically grown particles act directly as Cloud Condensation Nuclei (CCNs) and grow

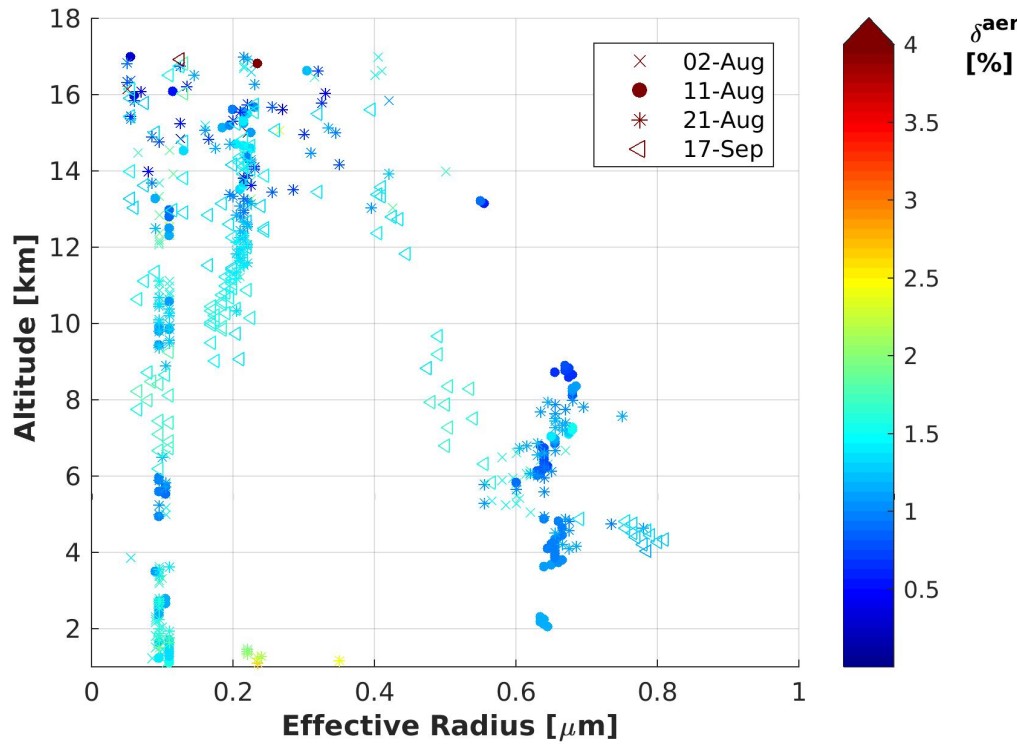

**Figure 8.** Averaged height-dependent estimation of the effective radius determined for during Lidar observations until 18 km altitude in correlation with the aerosol depolarization, $\delta^{aer}$, for all four observation days with KARL.

in the water phase, but do not serve as Ice Nucleating Particles (INPs). This can be seen in the low depolarization throughout the entire studied atmosphere. Furthermore, this process happened quickly, because otherwise the gap in effective radius in the troposphere between 0.2 $\mu$m to 0.5 $\mu$m would not exist. At an altitude of about 1 km, a distinct aerosol type appeared on 21 August. Only on this day at that height, the depolarization reached its maximum with 2.5 %, indicating slightly less spherical shape than in the layers above.

In the stratosphere (from 9–10 km) a different behavior is observed. A distinct aerosol signature appeared, characterized by an effective radius of approximately 0.2 $\mu$m, although dispersed values ranging from 0.2 to 0.4 $\mu$m are also observed, particularly between 12 and 17 km a.g.l.. When comparing the results from KARL with the retrievals from photometer observations (see Section 3.1.1), the effective radii are similar. It must be noted that generally the aerosol backscatter and extinction decreases with altitude (not shown here). Hence, in Figure 8 the lower altitudes needs to be "weighted" more than high altitudes

for a comparison with photometers. By comparison of Figures 3 and 8 it is obvious that the predominant mode around 0.1 $\mu$m (Accumulation) is tropospheric, while the second maximum (in Figure 3 weaker) around 0.2 $\mu$m is stratospheric. Particles > 0.5 $\mu$m seem to be less frequent in the lidar, compared to photometer. This is likely due to the selection effect that the lidar only

operates during clear sky conditions. Hence, clouds are underrepresented in the lidar data. Still, some hygroscopically grown aerosol might be visible in the lidar data which we relate to the range of effective radii between 0.55 to $0.8\ \mu m$. Further, the weak coarse mode with particles $> 1\mu m$ is missing in the lidar, as for this instrument we only assumed a one-modal aerosol distribution.

## 3.2 Origin identification

### 3.2.1 Columnar events CS1, C2 and C3

The footprint emission sensitivity together with the corresponding source contribution to BC and the continental contribution to BC at each model altitude for events CS1, C2 and C3 are depicted in Figure 9. These parameters have been integrated, for each event, by the temporal duration determined in the previous section. The source contribution panels in the middle column of Figure 9 indicate high concentrations of biomass burning BC during these events. Event CS1 (Figure 9, top row) is the one that presents the highest concentration, with maximum values of $140\ ng\,m^{-3}$ at 500 m a.g.l., but also a high concentration at 100 and 4000 m a.g.l. For this event the footprint emission sensitivity indicates that the BC was coming mainly from wildfires in central Siberia. The use of a continental mask indicated that the contribution of BC from fires in Russian territory was more than 95 % at heights between 0 and 4 km.a.g.l. (Figure 9).

Similarly, the highest BC concentrations during event C2 (Figure 9, middle row) were obtained at 4000 m a.g.l. However, modelled concentrations reached a maximum of only $40\ ng\,m^{-3}$. The source contribution to column BC shows that BC had two branches of origin, one from Siberia and another from North America. Wildfires contributed over 70 % of the BC mass arriving at the station at altitudes up to 4 km, and above 97 % of the mass was coming from Russia.

Finally, for event C3 (Figure 9, bottom row), the modelled BC concentrations at different heights reached a maximum at 4000 and 500 m a.g.l., with almost 70 and 60 $ng\,m^{-3}$ respectively. While the footprint emission sensitivity shows impact from Canada, the majority of the air masses arriving at the Zeppelin Observatory originated from central and western Siberia at 4000 m a.g.l. North American fires dominated only at lower altitudes of up to 500 m a.g.l. This last event was the second most intense observed during this summer, showing a good correlation with the AOD values observed in Section 3.1.1. In addition, both C2 and C3 showed a important contribution from North America. This agrees well with the hypothesis of the long range transport considered after observing a wide and uniform aerosol side distribution in Section 3.1.1 in these events, specially for C3.

As mentioned in the introduction, during the MOSAiC expedition a persistent 10 km deep aerosol layer in the UTLS, roughly from 7-8 km up to 17-18 km over the central Arctic, with clear a sign of smoke was observed. A layer around 10-15 km has also been observed in the data for summer 2019 analyzed here. Therefore, some lifting of the smoke must have taken place. The air in July-August 2019 originated from ongoing large wildfires over Siberia and low-wind and stagnant conditions allowed air to accumulate. The lack of evidence of strong pyrocumulonimbus (pyroCb) activity over these fires during the key period

in combination with CALIPSO smoke detections at 10 km, led Ohneiser et al. (2021) to invoke that self-lofting might be a possible mechanism resulting in the persistent UTLS smoke layer. In a more recent publication, Ohneiser et al. (2023) explicitly treat self-lofting as a credible alternative to pyroCb convection for raising large smoke masses from 2-6 km to the tropopause and cites the 2019 Siberian case and MOSAiC results as key evidence. In addition, (Tarshish and Romps, 2022) tried to answer
whether a dry firestorm plume (an intense conflagration that creates and sustains its own updraught wind system) can on its own reach the stratosphere. By using plume models (with and without entrainment), direct numerical simulations (DNS) and large-eddy simulations (LES) of idealized urban firestorms, they found that a dry plume starting at around 1 km (top of PBL) needs a temperature anomaly of about 60 K to stay positively buoyant up to a 15 km tropical tropopause. When they included entrainment, they found that for 1 km plume radius, mixing doubles temperature anomaly in the poles and sextuples it in the
tropics. They conclude that narrow and dry plumes need to be unrealistically hot to reach stratospheric heights. Then, they used DNS and LES to simulate realistic dry firestorms and found that they never get hot enough to reach the stratosphere staying at around 5 km, at maximum. When relative humidity in the plume increased above 50 %, pyroCb-like convection developed, which lifted fire plumes to tropopause or even to stratosphere. They conclude that even moderately moist environments allowed latent heating to push firestorm plumes to the stratosphere. Overall, whether the lifting of smoke in summer 2019 was due to
pyroCb-like latent heating (moist convection) (Tarshish and Romps, 2022) or due to radiative heating (self-lofting) (Ohneiser et al., 2021) requires further research. While plume-rise parametrizations with moist thermodynamics and pyro-convection are already in use by many global models (Ma et al., 2024; Ke et al., 2025), they are not relevant here, as FLEXPART used emissions from CAMS GFAS.

### 3.2.2 Intermediate periods with high AOD: from 14 July

During the periods between the aerosol events described above, relatively high AOD values have been also observed (AOD at 500 nm around 0.15-0.20 in July and August, dropping to 0.10-0.15 in September), which corresponded to fine aerosol with low absorption (see Section 3.1.1). Modelling BC levels calculated using the described emissions inventory did not show a clear source of the aerosol. At around that period, an unexpected series of blasts from a remote volcano in the Kuril Islands (Raikoke volcano) sent ash and volcanic gases streaming high over the North Pacific Ocean (Gorkavyi et al., 2021). The dormant period
ended around 18:00 UTC on 21 June 2019, when a vast plume of ash and volcanic gases shot up from its 700-meter-wide crater. Several satellites observed as a thick plume rose and then streamed eastwards (McKee et al., 2021; Kloss et al., 2021). Kloss et al. (2021) observed that afterwards, the plume separated into an ash-dominated component in the south and a $SO_2$-dominated component in the north. While the ash plume rapidly diluted and could not be further followed, the $SO_2$ plume persisted. On 23–27 June 2019 parts of the dispersed $SO_2$ cloud, which were observed with the Microwave Limb Sounder (MLS) satellite
data, presented heights between 11 and 18 km, with a peak concentration at 14 km (Gorkavyi et al., 2021).

Therefore, we further simulated the Raikoke eruption using a new inventory for the volcanic $SO_2$ emissions (Osborne et al., 2022). The vertical cross-sections of the modelled volcanic $SO_2$ arriving at Ny-Ålesund is illustrated in Figure 10. This figure shows that the volcanic aerosol reaches Ny-Ålesund specially from 14 July, when it was also observed an increase in the AOD

with respect to previous AOD values. This aerosol extended in a thick layer between 7 and 15 km a.g.l. for the first two weeks since its arrival, and afterwards it was mainly present between 10 and 15 km a.g.l., which is consistent with the observations made with CALIOP and KARL in Figure 6.

Whether the aerosols observed at such altitudes were ash or soot was under strong debate. Ansmann et al. (2021) reported that the aerosol was smoke misclassified as volcanic sulfate focusing on lidar measurements in Leipzig (Germany). The reason for the misclassification was the aerosol identification algorithm that assumes non-spherical smoke particles in the stratosphere. They reported that self-lofting particles reached the tropopause within 2–7 days after emission and finally entered the lower stratosphere as aged spherical smoke particles. These spherical particles were misclassified as liquid sulfate particles. As explained in the previous subsection, the same group hypothesized that the layer detected during the MOSAiC expedition mainly originated from extraordinarily intense and long-lasting wildfires in central and eastern Siberia in July and August 2019 which may have reached the tropopause layer by self-lofting (Ohneiser et al., 2021, 2023).

On the other hand, Boone et al. (2022) concluded that the stratospheric aerosol in the second half of 2019 over the Arctic consisted of Raikoke sulfate aerosol. This study was only based on observations of stratospheric infrared absorption spectra (in the framework of the satellite-based Atmospheric Chemistry Experiment mission). The authors found no indication of the presence of smoke. Gorkavyi et al. (2021) reported a high $SO_2$ concentration and aerosol clouds using data from the Ozone Mapping and Profiler Suite sensors on the Suomi National Polar-orbiting Partnership satellite. However, a recent comment on this publication made by Ansmann et al. (2024) states that the AOD derived from the measurements used in this analysis was four times higher than the corresponding AOD for the precited $SO_2$ concentration from the Raikoke volcano. In this review, it is concluded that the Arctic UTLS aerosol was likely dominated by smoke in the lower part and by sulfate aerosol in the upper part. Kloss et al. (2021) combined both observations and model simulations of the Raikoke eruption and found that a few days after the eruption, the volcanic plume was entrained in the Aleutian cyclone, and a month after, it circled the Earth (it reached Europe around 1 July). Accordingly, stratospheric AOD (sAOD) was as high as 0.045 (at 449 nm) at higher north-hemispheric latitudes, with an average value of 0.025 at longer wavelength (visible, 675 nm), while the background AOD was still enhanced in the North Hemisphere one year after the eruption. Finally, Vaughan et al. (2021) used a Raman lidar in the UK and found a thin layer at 14 km on 3 July, with the first detection of the main aerosol cloud on 13 July, which agrees very well with our own observations in Ny-Ålesund. At that period the aerosol was confined below 16 km extending to 20 km. The authors further reported a sustained period of clearly enhanced AOD from early August, with a maximum value (at 355 nm) around 0.05 in mid-August and remaining above 0.02 until early November.

### 3.2.3 Surface events S1 and S2

Finally it has been investigated the origin of the surface events S1 and S2. The footprint emission sensitivity computed for the day of each event and the corresponding temporal evolution of the source and continental contributions to BC for the days of the events and days around have been plotted in Figure 11. The footprint emission for events S1 and S2 (Figure 11,

left column), indicates that S1 had its source in Central Europe while, in S2, the BC is coming from Eastern Europe. This is
corroborated by the continental contribution (Figure 11, right column), which shows a main contribution from Europe together
with a smaller contribution from Russia, more evident in S2. Both events had an anthropogenic source, mainly due to trans-
portation (TRA) and commercial and domestic (DOM) emissions (see Figure 11, middle column). The products for the age of
the aerosol reaching Svalbard indicate that, in general, aerosol in these events traveled faster compared to CS1 (See Figure S5
in the supplement), which agrees with the slightly lower $SSA_{530}$ values observed.

By the late summer and early autumn, aerosol travel from lower latitudes to the Arctic is not as efficiently removed along
the pathways as earlier, leading to a more efficient transport of polluted air masses and therefore an increase of the aerosol
scattering and absorption coefficients measured (Gilardoni et al., 2023; Garrett et al., 2011). In addition, Pulimeno et al. (2024)
showed an evident change of atmospheric transport regime in the summer of 2019, from North America during the summer
period to North Europe and Russia during autumn and early-winter, which supports the anthropogenic origin of the aerosol
measured during these events.

### 3.3 Aerosol radiative impact

Figure 12 presents the times series of anomaly of DNI ($\Delta_{DNI}$) from June to September 2019 calculated as described in
Section 3.3; only the cloud-free cases are used, selected depending on the value of the diffuse ratio (DifR) described. The
adjacent years, 2018 and 2020, are also shown.

Despite the large amount of data with cloudy conditions (gaps in temporal series in Figure 12), all cloud-free cases presented
negative $\Delta_{DNI}$ values in 2019 since 5 July, which can be explained by a smaller amount of DNI in 2019 due to the high
aerosol load. Compared to the previous and next year, the $\Delta_{DNI}$ average in July-August 2019 was more than four times larger
(negative sign), with a mean value of $-73.6$ W m$^{-2}$. In September, anomalies close to zero are observed, likely due to the
reduced radiation received towards the end of the light season. The large standard deviation observed shows the complexity of
this analysis, with multiple conditions (mainly variation in aerosols and clouds) sometimes playing roles in opposite directions.
However, in general, the negative sign of $\Delta_{DNI}$ is a good proxy for the effect of the decrease in the direct component of solar
radiation.

### 4 Conclusions

In order to understand the events leading to the high aerosol optical depth (AOD) and back-scatter values observed during the
summer of 2019 (June to September), all available observational data in Spitsbergen, together with FLEXPART calculations,
have been combined in this study.

The present analysis revealed that the aerosols in Svalbard in the summer of 2019 reached the troposphere and also the
stratosphere. However, these periods of high AOD were perceived differently by the remote sensing and in-situ instruments. In
the remote sensing instruments (atmospheric column) the whole summer from mid July till end of September showed slightly

variable but continuous unusual high AOD. Three events in the atmospheric column were identified: 6 to 11 July, CS1, 26 to 30 July, C2, and 6 to 19 August, C3. Contrary, the in-situ stations records were not remarkable except for three surface events: 5 to 9 July, CS1, 30 August, S1, and 12 September, S2, which showed high aerosol scattering and absorption coefficients at surface level. Only the first event (CS1) was observed simultaneously at the surface and at elevations up to 4000, suggesting that most of the aerosol during this summer was advected above the boundary layer. Due to cloudy conditions, remote sensing observations were not possible during S1 and S2. Therefore, column-integrated measurements are not representative of surface conditions, and they may miss some surface pollution events.

The correlation of aerosol absorption and scattering coefficients between the two nearby in-situ sites used for the study was relatively weak (Pearson coefficient of 0.71 and 0.34, respectively), indicating that the deposition of aerosol in Ny-Ålesund might be dominated by local micrometeorological processes. Therefore it is useful to considerate both stations for comparison with climate models and long-range transport.

The aerosol properties from sun-photometer and lidar fairly agrees. Accumulation mode particles (around $0.1\mu m$) were found mostly in the troposphere, and a weaker Accumulation mode of larger particles (around $0.2 \mu m$) was observed mainly in the stratosphere. This second maximum was not found in the aerosol size distributions from mid-July when the first particles from the Raikoke volcano arrived to Ny-Ålesund. This might indicate the intrusion of some particles in the stratosphere, however, no lidar measurements were available in July in order to corroborate it. Some big particles, which may had grown hygroscopically over water (as they maintained their weak depolarization), were also observed. Coarse mode particles were very sparse, as expected after long range transport.

The backward analysis of airmasses using FLEXPART has been useful to shed light on the origin of the aerosol, identifying three different sources: Raikoke volcano, biomass burning (BB) events and anthropogenic pollution. The modelled BC concentrations agrees with AOD values measured with the sun-photometer, correctly determining the intensity of each event. FLEXPART simulations indicated that the Raikoke volcano contributed until end of August to the AOD; however the remote sensing measurements showed a similar pattern in September than in the previous months, pointing out that probably remnants of the volcanic particles and also BB events stayed longer in the stratosphere. Further, different BB sources and different anthropogenic pollution sources were detected. Though the BB events occurred mainly between 500 and 4000 km a.g.l., some self-lofting might have occurred so that it reached the stratosphere, probably after even C2. The anticyclonic system observed in Siberia (Antokhina et al., 2023) likely enhanced the transport of aerosol to the Arctic, first northwards into the Arctic and then eastwards towards North America. Hence we may have seen the rest of this mixed smoke. These mechanisms suggest that the BB contribution was likely more important than the volcano contribution in the upper troposphere - lower stratosphere (UTSL). Over the period from June to September, in the surface in-situ samplers, the BB and volcanic impact was minor, while anthropogenic emissions had more influence.

Despite its variable origin, this long-lasting summer episode of high AOD, consisted of spherical (low depolarization), weakly absorbing particles (high SSA), and small particles (Accumulation mode). As it occurred in late summer (relatively high solar altitude and low surface albedo), it contributed to a clear cooling signal; the anomaly of solar direct normal irradiance ($\Delta_{DNI}$) was generally below -50 W/m$^2$, with a mean of -73.6 W/m$^2$ in July and August.

The analyses of the results obtained from various methods and climate models was not a trivial task. It is obvious that there is a strong need for dedicated campaigns to bring together all methods of AOD studies, including both the in situ and remote sensing ones. While methods, techniques and instruments are already available since decades, well designed actions have never been conducted in the Arctic. Ship borne studies with in situ aerosol and Microtops II measurements facilitate very good study ground, especially, due to the fact that the ocean boundary layer is less disrupted than that over land, especially in the fjords, where aerosol studies can be affected by mountain orography (e.g. in Ny-Ålesund). In general, for the complete understanding of this long-lasting high AOD episode during summer 2019, the combination of the different instruments in and around Ny-Ålesund has been crucial, highlighting the importance of multi-instrumental studies and collaborations between different institutions and research areas.

*Data availability.* The AOD measurements can be found in Rehearsol https://doi.org/10.21343/1gaj-4645. Aerosol inversion properties can be downloaded at AERONET webpage: https://aeronet.gsfc.nasa.gov/. In-situ data for GAL associated with project RIS ID 3693: Gruvebadet Atmospheric Laboratory available under request to the authors In-situ data for ZEP is available under request to the authors. CALIOP data can be found in http://www.icare.univ-lille1.fr/ KARL data is available under request to the authors. The radiation data can be found at Maturilli (2020) https://doi.org/10.1594/PANGAEA.914927. The results of the FLEXPART model simulations are openly available via https://atmo-access.nilu.no/ZEPP_lidar.py.

*Author contributions.* SHA and SaG conceived the main ideas of the study, developed the core concepts, and wrote the manuscript with input from all authors. SHA, DM, and RR contributed to the columnar aerosol measurements and their analysis. NK, SK, and TZ also contributed to the columnar aerosol measurements. SG, MM, DHR, and RK provided and processed the in-situ data. KS was responsible for the satellite data. SaG and CR developed the vertically-resolved aerosol properties analysis and handled the KARL data. NE and SE performed the FLEXPART simulations and conducted the modelling analysis. DM contributed to the radiative impact assessment. SHA, RR, DM, and SK secured funding for the study. All authors participated in scientific discussions and reviewed the manuscript

*Competing interests.* At least one of the (co-)authors is a member of the editorial board of Atmospheric Chemistry and Physics.

*Acknowledgements.* FLEXPART model simulations are cross-atmospheric research infrastructure services provided by ATMO-ACCESS (EU grant agreement No 101008004). The simulations were performed on resources provided by Sigma2 - the National Infrastructure for High Performance Computing and Data Storage in Norway. We thank NASA and CNES engineers and scientists for making CALIOP data available. We thank Piotr Sobolewski, Piotr Glowacki, Grzegorz Karasinski, Pawan Gupta, Elena Lind and their staff for establishing and maintaining the Hornsund site from AERONET used in this investigation. This work is part of the project TED2021-131211B-I00375 funded by MCIN/AEI/10.13039/501100011033 and European Union, "NextGenerationEU"/PRTR. This work was supported by the Ministerio de Ciencia e Innovacion (MICINN), with the grant no. PID2021-127588OB-I00 and is based on work from COST Action CA21119

HARMONIA. Financial support of the Department of Education, Junta de Castilla y León, and FEDER Funds is gratefully acknowledged (Reference: CLU-2023-1-05). The authors acknowledge the support of the Spanish Ministry for Science and Innovation to ACTRIS ERIC. Grant PID2022-142708NA-I00 funded by MCIN/AEI/10.13039/501100011033 is also acknowledged. We also want to thank the Marie Sklodowska-Curie Staff Exchange Actions with the project GRASP-SYNERGY (grant no. 101131631). The authors thank CNR and the staff of Dirigibile Italia for their support managing the Gruvebadet instrumentation. The authors would like to thank PAIGE (Italian-German

Partnership) project. K.S. was supported by the MIT3D CCN-1 SVAR project (ESA 4000135992/21/I-DT-lr810 CCN-1) and S.E received funding from the European Union's Horizon Europe research and innovation program (101081395; EYE-CLIMA). Observations at Zeppelin observatory are Supported by Swedish Environmental Protection Agency (Naturvårdsverket), ACTRIS-Sweden and Norwegian polar Research Institute (NPI).

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

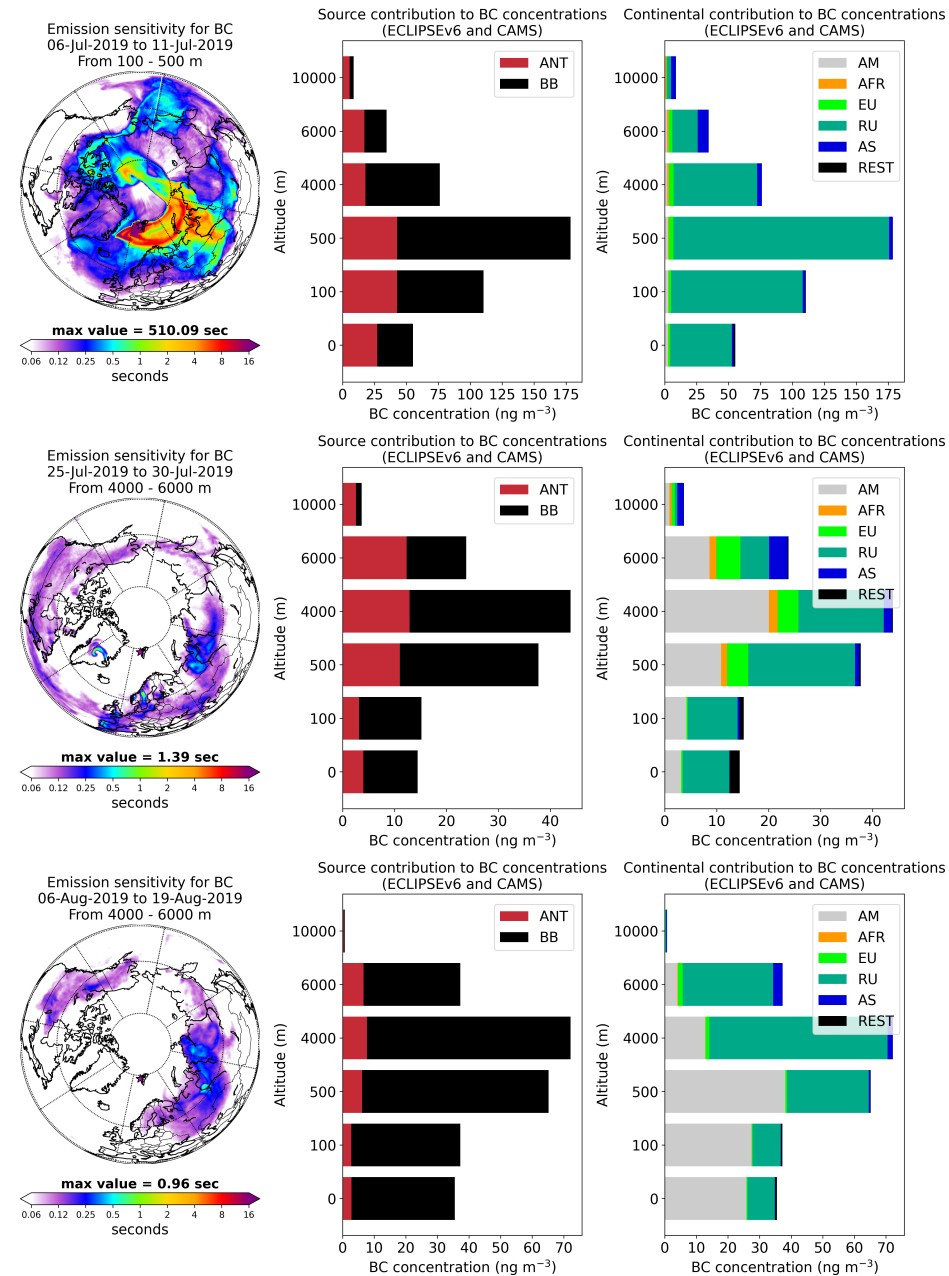

**Figure 9.** FLEXPART backward modelling results for the source regions of the airmasses arriving at ZEP Observatory: in top row from 6 to 11 July (CS1), in middle row from 26 to 30 July (C2) and in bottom row from 6 to 19 August (C3). On the right column it is shown the emission sensitivity for BC computed for the days of the events. The release height for the footprint is determined based on the altitude of maximum BC concentrations and is provided in the header. The modelled BC concentrations at five height levels (lower boundary: 0 m, 500 m, 4000 m, 6000 m and 10000 m) colour-coded by source and continental contributions, are shown in the middle and right columns respectively. With respect to the source we have distinguished between biomass burning (BB) and anthropogenic (ANT). For the region of origin, the labels used are: America (AM), Africa (AFR), Europe (EU), Russia (RU), Asia (AS) and rest of the world (REST).

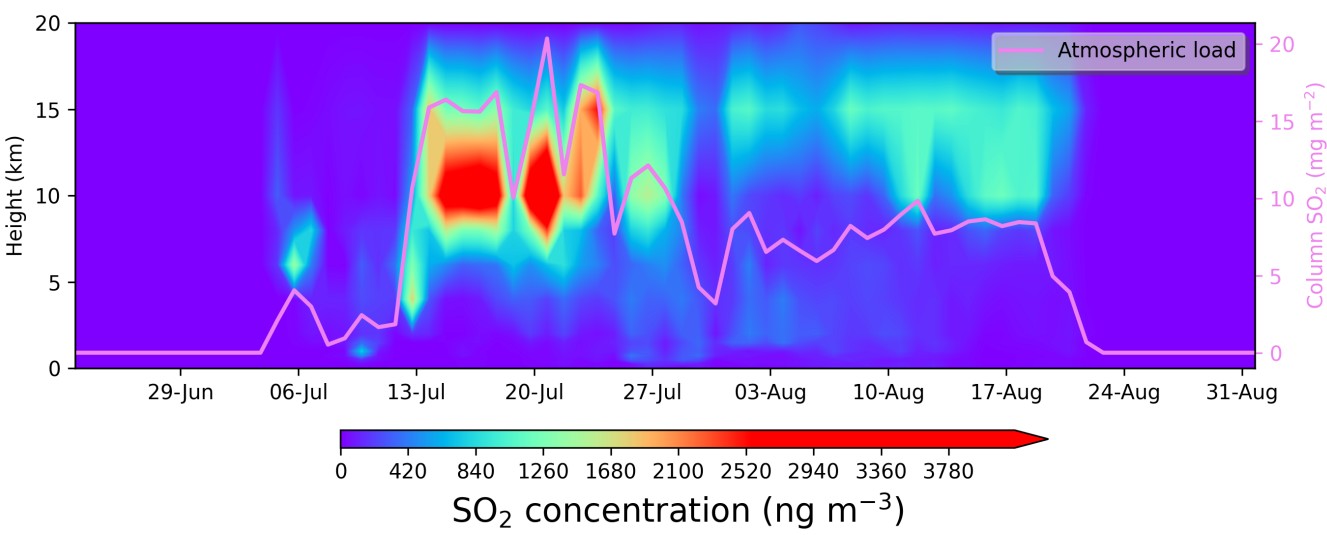

**Figure 10.** Time-height section of the volcanic SO$_2$ concentration (coloured contours) as modelled by FLEXPART on its arrival at Ny-Ålesund station from end of June to end of August 2019. The magenta line indicates the modelled total column concentration.

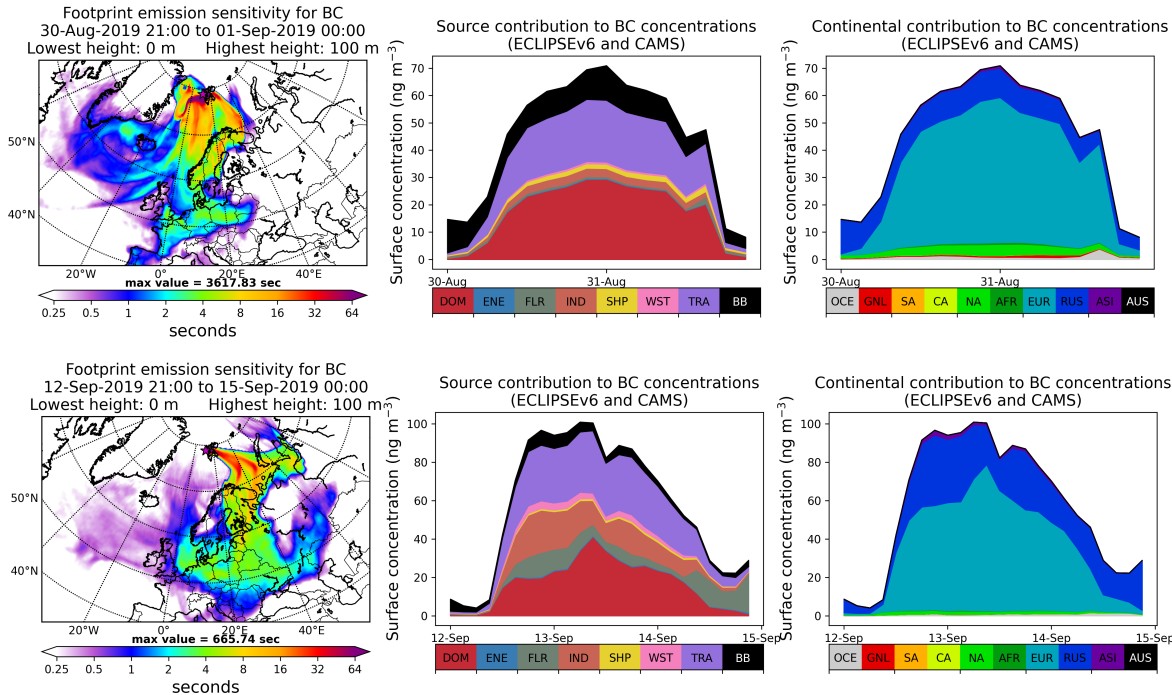

**Figure 11.** FLEXPART backward modelling results for the source regions of the airmasses arriving at ZEP Observatory during the enhanced pollution events on 30 August (S1, top row) and on 12 September (S2, bottom row). On the right column it is shown the emission sensitivity for BC computed for the days of the events. The temporal evolution of the modelled BC concentrations at the surface during the days of the events and days around, colour-coded by source and continental contributions, are shown in the middle and right columns respectively. With respect to the sources it has been distinguished between the biomass burning (BB) emissions and the following anthropogenic emissions: residential and commercial (DOM), energy production (ENE), gas flaring (FLR), industry (IND), shipping activities (SHP), waste treatment and disposal sector (WST) and transportation (TRA). For the region of origin the labels used are: America (AM), Africa (AF), Europe (EU), Russia (RU), Asia (AS) and rest of the world (REST).

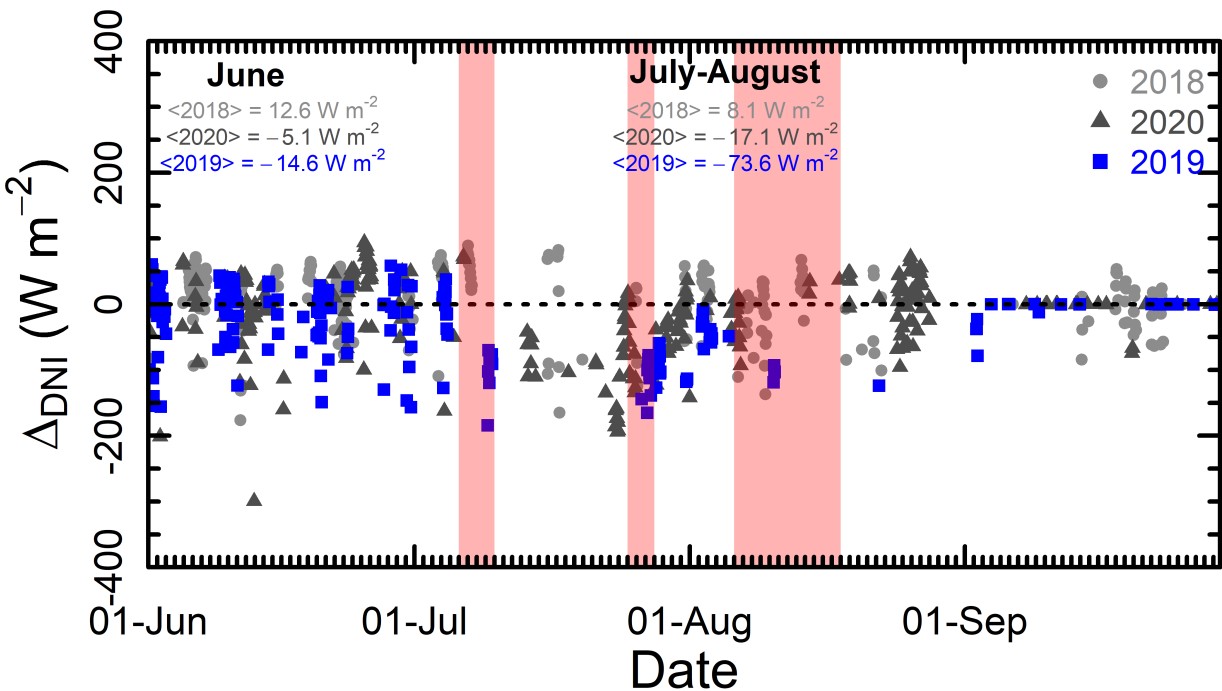

**Figure 12.** Anomaly of solar direct radiation in 2019, 2018 and 2020 with respect to the reference value for the 2006-2020 period as described in Equation 2. The average anomaly of each year has been included in the left corner. The red shaded areas highlight the three columnar events analyzed throughout the study.