# Peer review of "Exceptional high AOD over Svalbard in Summer 2019: A multi-instrumental approach"

_EGUsphere, 2025_

## Author Comment (AC1)

**Response to Dipesh Rupakheti Referee comments for the manuscript "Exceptional high AOD over Svalbard in Summer 2019: A multi-instrumental approach" by Sara Herrero-Anta et al. in AMT**

*AC: First of all, we would like to thank the time and effort of the referee for their review of the manuscript. Reviewer comments (RC) are in bold font and author comments (AC) in italic font.*

**RC: This manuscript attracted my attention as I have investigated the columnar aerosol properties utilizing AERONET datasets over another important region (South and Central Asia). I have provided some suggestions to consider while revising this work:**

**RC: L19: 'than the globe' reads awkward; revise.**

*AC: This phrase is literally the title of the known article 'The Arctic has warmed nearly four times faster than the globe since 1979' by Rantanen et al. (2022).*

*We have changed it to 'nearly four times faster than the rest of the globe' in the new version of the manuscript.*

**RC: L23: reword 'present'.**

*AC: This has been changed to 'show' in the new version of the manuscript.*

**RC: Quantitative information based on relevant earlier studies (already cited) must be included in the Introduction section.**

*AC: Thank you for the remark. Due to the variability of the methodology of studies conducted in the Arctic where, well designed actions have rarely been conducted, we decided it was better to give a qualitative information in the introduction. However we do give quantitative and relevant information for our study about the results obtained in the MOSAIC campaign and Pulimeno et al. (2024) studies. Relevant information is also given during the analysis of FLEXPART results, but again, due to the variability of the methods we preferred to give the qualitative information.*

**RC: Figure 1: What do different colors indicate? Please elaborate on the abbreviations in the figure caption.**

*AC: This figure caption in the new version of the manuscript indicates the following:*

*"Topographic map of the vicinity of Ny-Alesund and its location on Svalbard. The main stations used for the study have been located in the map: Zeppelin Observatory (ZEP), Gruvebadet Atmospheric Laboratory (GAL), AWIPEV and Sverdrup. The colors indicate*

*the altitude; blue indicates water surface. Map created using the dataset by Moholdt et al. (2019).''*

**RC: L91: State the relationship between AE value and particle size.**

*AC: The relationship between AE and particle size distribution is not straightforward and nor is the focus of this study. One may refer to specific literature to consult this relationship. We have included a citation for that in the new version of the manuscript (Kokhanovsky, 2008):*

*"AE is related to the aerosol particle size (Kokhanovsky, 2008). A mean value of AE equal to 1.3 is observed for the average continental aerosol (Ångström, 1929), while values close to 0 indicate coarse particles."*

**RC: L111: Which data level was used for AOD and AE retrieved from the AERONET website? This is very important regarding QA/QC.**

*AC: We use the same data level for AOD and AE as for inversion products. This might not be clear in the old version of the manuscript. For the new version we have changed it:*

*"All AERONET data used correspond to level 1.5 products (version 3)."*

**RC: Figure 2 caption: shaded box color is not red (at least to me).**

*AC: Thank you for the remark. This has been corrected in the new version of the manuscript:*

*"The time periods highlighted in dark red, pink and red shaded areas correspond to the three events identified as aerosol events in the column, respectively: 6-10 July (C1), 25-28 July (C2) and 6-17 August (C3)."*

**RC: L247: rephrase 'collects'.**

*AC: This has been exchanged by 'summarizes'.*

**RC: L259: cite reference for longer transport time in August.**

*AC: This is only a hypothesis for the reason of the different size distributions in July and August, not a general behaviour for the times of transport to the Arctic. In order to make this more clear we have slightly modified the sentence: 'might had been longer in August compared to July'.*

**RC: Figure 3: My suggestion is to plot event-average values here and move the detail figure to supplementary, as the present figure looks crowded with hard-to-decipher information.**

*AC: We understand it might be a bit crowded. However, we believe it is more useful to look at the individual retrievals, since we might lose information when conducting the average. We already give the averaged values of all the available aerosol properties from the sun-photometer in Table 2.*

**RC: L297: Those lower values refer to instantaneous values?**

*AC: Yes, this has updated in the new version: For event CS1 the mean $SSA_{530}$ was 0.95 ±0.01 with lowest instantaneous values equal to 0.92.*

**RC: L303: With respect to GAL?**

*AC: This was changed to 'Regarding GAL observations' in the new version of the manuscript.*

**RC: Figure 12: In the x-axis, correct the spelling for August.**

*AC: Thank you for the remark, this has been corrected in the new version.*

**RC: L458: As a result…. This sentence could be removed.**

*AC: Thank you for the remark, this sentence has been removed in the new version of the manuscript.*

**RC: L462: I don't think such detailed information on the aerosol event occurrence date is required, at least here.**

*AC: Thank you for the remark. As we are referring to the different aerosol events, we believe it is helpful to state the dates, instead of only saying CS1, C2…*

**RC: Conclusion section: The current version reads like a simple summary of each subsection, which needs revision.**

*AC: The conclusions section has been updated in the new version of the manuscript according to the comments received from the different reviewers.*

---

## Author Comment (AC2)

**Response to the Anonymous Referee #1 comments for the manuscript "Exceptional high AOD over Svalbard in Summer 2019: A multi-instrumental approach" by Sara Herrero-Anta et al. in AMT**

*AC: First of all, we would like to thank the time and effort of the referee for their review of the manuscript. Reviewer comments (RC) are in bold font and author comments (AC) in italic font.*

Author's answer to Anonymous Referee #1

**RC: General comment: The authors analysed Arctic aerosol observations and discuss the findings. Pollution source identification based on FLEXPART modelling is part of the study. The focus is on the summer of 2019. Wildfire smoke, anthropogenic pollution as well as volcanic sulfate aerosol (originating from the Raikoke eruption) polluted the troposphere and lower stratosphere from the surface up to about 20km height. The manuscript contains interesting information and is clearly worthwhile to be published in ACP. However, many questions came up during reading and need to be clarified as part of the revision of the paper. Major revisions are needed.**

**Detailed comments and questions:**

**RC: Line 9: please state clearly: do you mean diameter of radius? …. 0.1-0.2 micrometer. Accumulation mode particles cover the radius size spectrum from about 100 to 500 or even 1000 nm! What do you mean with 0.1-0.2 micrometer?**

*AC: We refer to the effective radii observed during the analysis. It was not correctly mentioned. It has been specified in the new version of the manuscript:*

*We replaced: "(0.1-0.2 μm)" with "(with effective radii between 0.1 and 0.2 μm)"*

**RC: Line 23: please be more precise: Do you mean the boundary layer or the free troposphere. Aerosols in the free troposphere are usually related to long-range transport, and not local aerosol production.**

*AC: In the paragraph starting at line 23, we provide an overview of the main aerosol sources in the Arctic. We do not refer to aerosols in specific vertical layers; rather, we indicate that in summer aerosol particles are mainly of local origin, mainly from new particle formation. Long range transport occurs more sporadically and is often confined to specific events.*

*This sentence has been added to the paragraph: "Long range transport occurs more sporadically and is often confined to specific events."*

**RC: Lines 25-28: Besides the given references one needs to mention recent observations from MOSAiC (Ohneiser et al., ACP, 2021, Ansmann et al., ACP 2023) and also the satellite observations presented by Kloss et al. (ACP, 2021).**

*AC: As stated in the previous comment, this paragraph gives an overview of the aerosol sources in the Arctic not an overview of the observations in the Arctic. The MOSAiC campaign (Ohneiser et al., ACP, 2021) was introduced in line 48, when we presented the measurements in the Arctic. Ansmann et al. (2023) was not included, so it is included now in the new version of the manuscript:*

*"In particular, an important expedition took place from September 2019 to October 2020: MOSAiC (Multidisciplinary drifting Observatory for the Study of Arctic Climate; Shupe et al., 2022), the largest Arctic field campaign ever conducted, which , among other data, provided an annual cycle of aerosol properties over the central Arctic (Ansmann et al., 2023)."*

*The paper from Kloss et at. (2021) was also included in the document, but it has been also added in this lines:*

*"...and in elevated atmospheric layers (Cheremisin et al., 2019; Kloss et al., 2021)."*

**RC: Line 30: What about indirect aerosol effects, i.e., the impact on water cloud, mixed-phase cloud and cirrus formation, and related precipitation processes.**

*AC: We have included the following:*

*Aerosol particles also have indirect effects due to their capacity to act as cloud or ice condensation nuclei, thus affecting clouds properties and formation, and the hydrological cycle, among others (see Lohmann and Feichter, 2005).*

**RC: Line 51: Please use lofting instead of lifting throughout the article!**

*AC: Thank you for the remark, this has been corrected in the new version of the manuscript.*

**RC: Line 53-55: The arguments show already that in situ observations at ground are not just helpful in the study of the aerosol conditions in the entire vertical column. Especially removal processes by washout events permanently clean the lowermost 200 m of the Arctic troposphere so that surface observations cannot be used to describe the cloud- and radiation-relevant aerosol conditions in the Arctic. Such statements should be included in the article. Furthermore, how is the 2% contribution by biomass burning identified? If this finding is based on BC information, the conclusion may be wrong. Wildfire smoke consists to 95-98% of organic carbon (OC).**

*AC: The study by Pulimeno et al. (2024) presents a novel method to identify biomass-burning events based on optical measurements combined with chemical analyses and air-*

mass back trajectories, using positive matrix factorization. In this section, we aim to introduce recent studies conducted in the Arctic. I was unable to find any work stating that "removal processes, particularly washout events, permanently clean the lowermost 200 m of the Arctic troposphere". However, we have addressed this issue in the revised version of the manuscript by including the following statement:

"Due to the variability of the residence times, removal processes and transport of aerosols in the Arctic, the aerosol contribution in the boundary layer and in the free troposphere is very different (Willis et al., 2018b; Yao et al., 2023). Therefore, surface observations are representative for boundary layer conditions but may not be sufficient to characterize the entire atmospheric column."

**RC: Line 81: mention season and year of the R/V OCEANIA field studies!**

*AC: This line has been updated to include season and year.*

*The phrase: "In addition to the ground stations, data recorded onboard the research vessel R/V OCEANIA, which was travelling through the Fram strait during the period of study, has been used."*

*Has been changed to this: "In addition to the ground stations, data recorded in summer 2019 onboard the research vessel (R/V) OCEANIA from the Polish Academy of Sciences, has been used. The R/V OCEANIA conducts regular measurement campaigns in the Arctic, and it was travelling through the Fram strait during the period of study."*

**RC: Line 135: KARL seems to be a very powerful lidar (50 laser pulses per second, about 200mJ per pulse at 355, 532, 1064nm, 70 cm telescope)! Why are almost no lidar observations shown? I expected particle extinction and smoke lidar ratio profiles at several wavelengths! Some Raman lidar applications! But only a few low-quality color plots are presented together with not trustworthy inversion products without showing any basic multiwavelength lidar observations. This is not good, and should be improved! I will come back to this point in more detail later on in this review.**

*AC: Thank you for your remark and interest on the lidar data! We will generally give more information on this in the new version of the manuscript (see following comments). Unfortunately for summer 2019 our data coverage is very poor. This is due to 2 reasons: the lidar is not switched on in cloudy conditions, as the return from cloud bottom may lead to saturation and harms the photomultipliers. Second, current safety regulations require the presence of a skilled lidar operator on site. Therefore, unfortunately, we do not have more days than presented in this manuscript for summer 2019. Some days for fall have been published in the Ohneiser 2021 paper.*

*Raman data: please note that due to the altitude of the events and polar day conditions the extinction from the N2 channels are poor. Beyond 9km the lidar profiles at 387nm and 607nm are basically noise. We cannot discuss the Raman channels in this work.*

**RC: How are the aerosol backscatter coefficients computed? I assume by using the Fernald method!**

*AC: Yes, we use a method similar to Klett 1985 and add as a quote Speidel and Vogelmann 2023. We have included this in the new version of the manuscript (see following comments). https://opg.optica.org/ao/fulltext.cfm?uri=ao-62-4-861*

**RC: What particle lidar ratios are assumed in the Fernald data analysis at the different wavelengths? In the case of aged wildfire smoke, the lidar ratios are about 55 sr (at 355nm), 85sr (at 532 nm) according to the report of Ohneiser et al. (2021) and about 100 sr (at 1064nm) for aged smoke as described in other papers. Is such a lidar ratio spectrum considered? In the case of sulfate aerosol (Raikoke) a similar lidar ratio spectrum holds but with lower lidar ratios, maybe 35, 60, and 75 sr. All this needs to be mentioned. The basic lidar products are the backscatter coefficient spectra and they are influenced by the lidar ratio input values.**

*AC: Right. For the stratosphere we used LRs of 70sr / 45sr / 45sr for 355nm, 532nm and 1064nm respectively. We have included this in the new version of the manuscript (see following comments). Of course, we carefully tested several lidar ratios. For 532nm, high LR >= 60 sr seem unlikely because otherwise the backscatter at the tropopause in clear conditions becomes very low. Just for your information we show the solution for 2ⁿᵈ Aug, 532nm with low and high LR. You can see that the backscatter only weakly depends on the assumed LR. For this reason (very low extinction at 1064nm) the LR in the infrared wavelength does not matter (and cannot easily be constrained neither). Even for 532nm the dependence of backscatter on the LR is <= 10% Tropopause "worst region" changing LR from 45sr to 80sr.*

[Figure]

**RC: Is the use of simply three backscatter coefficients really sufficient to retrieve the effective radius of the particles? I would not trust these results, especially not in the case of different aerosol types above each other which may be even partly mixed in the UTLS height range.**

*AC: Right. You refer to the Veselovskii paper from 2002. If one does an inversion of lidar data backscatter and extinction information is needed. As we can not do this, we need to work with further simplifying assumptions (line 150 of old version): we estimate a priori the refractive index and we restrict to a one-modal log-norm distribution. According to our previous work (e.g. Böckmann [https://doi.org/10.3390/rs16091576](https://doi.org/10.3390/rs16091576)) the choice of the refractive index will probably not critically affect the solution. We have repeated our assumptions at presenting Fig. 8:*

*"The daily averaged backscatter profiles have been used to estimate the height-dependent effective radius of the aerosol for each day, as described in Section 2.2.3: an a priori refractive index and a one-modal log-norm distribution are considered."*

*We have also included the following in Section 2.2.3:*

*"According to the previous work of Böckman et al. (2024) the choice of the refractive index will probably not critically affect the solution."*

**RC: Lines 185-189: CALIOP extinction profiles are used. When using the CALIOP elastic-backscatter lidar profiles, again a lidar ratio has to be assumed to obtain the backscatter and extinction profiles! I guess, the CALIOP science team used 70 sr for smoke and about 40 sr for sulfate particles. Please provide numbers here. The uncertainty in the products are high, higher than 50% (in terms of relative errors), I speculate!**

*AC: Thanks for the very relevant comments. We have added a figure (Figure S3, see following comment) to the Supplementary, which shows the classification (read out from the vertical feature masks of the profile closet to Ny-Ålesund) and a table (Table S1) which indicates the lidar ratios, used by the CALIOP retrieval teams for conversion of the measured backscatter into the estimated extinction. For the stratosphere layer 50 sr ± 18 sr is used for sulfate and unclassified layers, and 70 sr ± 16 sr for smoke.*

*Yes, we agree that there can be larger errors associated with the extinction profiles shown, e.g., the LR for sulfate/unclassified has an uncertainty of 36%. In addition, one can't fully exclude misclassification of the aerosol-type and the associated systematic uncertainty. To keep the visibility, we do not include error estimates in the extinction profiles, but have added given uncertainties to the tropospheric and stratospheric AOD estimates.*

*This has been included in Section 2.2.5 of the new version of the manuscript:*

*"The classification (read out from the vertical feature masks of the profile) and the lidar ratios, which are used by the CALIOP retrieval teams for conversion of the measured backscatter into the estimated extinction have been included in Table S1 in the supplement."*

*And in Section 3.1.3 we have included the following:*

*"The equivalent Figure S3 in the supplement indicates the classification of the layer aerosol type."*

*Table S1: Lidar ratios S532 (sr) for tropospheric and stratospheric aerosol subtypes in V4 (see Kim et al., 2018) and in V4.5 (see Tackett et al., 2023), respectively.*

| Tropospheric aerosol subtypes (V4) | |
|---|---|
| Dust | $44 \pm 9$ |
| Polluted continental/smoke | $70 \pm 25$ |
| Polluted dust | $55 \pm 22$ |
| Elevated smoke | $70 \pm 16$ |
| Stratospheric aerosol subtypes (V4.5) | |
| Sulfate | $50 \pm 18$ |
| Smoke | $70 \pm 16$ |
| Unclassified | $50 \pm 18$ |

**RC: Line 201: To my opinion, more information (some kind of a general overview and introduction) on the optical properties of smoke particles is needed. The BC content of aged smoke is 2-3% as can be found meanwhile in many modelling papers (references may be found in Ohneiser et al., 2023). Smoke particles mainly consist of OC (95-98%). Particle density of smoke particles is roughly 0.9-1.3 g/cm3. The OC content contributes to self-lofting because of the ability to significantly absorb even at wavelengths greater than 400 to 500 nm. In the case of a smoke AOD of 2-3 at 500 nm, self-lofting leads to ascent rates of 3 km per day. Ascents rates of 500 m per day are still possible for AODs of the order of 0.5-1. In the case of 30 days of transport, pronounced smoke layers may thus ascent, on average by 100 to 200 m per day, and in the beginning (shortly after emission) when the smoke plumes are optically dense, on average by 500 to 1000 m per day. Even if self-lofting is not considered in the FLEXPART simulations, a discussion on the consequences is needed. All in all, section 2.2.6 must be updated by considering self-lofting aspects.**

*AC: This is a very interesting point. We have now included some information about self-lofting in Section 2.2.6:*

*"These components can absorb radiation, warming the surrounding air and inducing upward motion that lifts the aerosol (Johnson and Haywood, 2023); i.e., the so-called self-lofting mechanism. GFAS uses two different models to calculate the injection height, based on satellite observed FRP and ECMWF forecasts of key atmospheric parameters (Rémy et al., 2017). Radiative self-lofting in global models, such as FLEXPART, is not considered yet, but the scientific basis now exists with the ECMWF radiation scheme (ECRAD) that computes shortwave heating rates of an imposed smoke layer (Ohneiser et al., 2023). However, online implementation of this module in global models might be demanding, due to the need of remote sensing data as input parameters (e.g., CALIOP aerosol observations, MODIS aerosol optical depth retrievals etc.). A more detailed*

*discussion of the potential mechanisms responsible for self-lofting is provided in the FLEXPART results section (see Section 3.2.1)."*

*Since self-lofting is not represented in the model nor we have specific data available to do this, we believe it is not appropriate at this stage to provide an educated estimate of how many meters per day the smoke may have ascended.*

*However, we have included a discussion on what is known in modelling perspectives about self-lofting and pyroCb-like convection implemented in global models, such as FLEXPART. We believe this is more helpful, however if the Reviewer insists that injecting mass higher than stated in the CAMS GFAS emissions would help here (even though this is tuning of the model), we are willing to do it in a next review round.*

**RC: Line 235: Why do you not use the period from 2002-2018 as reference?**

*AC: Since the dataset is available until 2020 we include 2020 to have a more representative reference.*

**RC: Line 245: The Siberian fire episode from the beginning of July 2019 to mid of August 2019 was already discussed in Ohneiser et al. (ACP, 2021 and 2023).**

*AC: We are not discussing the Siberian fires in that line, only showing the AOD measurements.*

**RC: Line 251: … when the peaks oscillate from 0.01 to 0.04 …. Please explain precisely: what do you mean here?**

*AC: with peaks we were referring to the maximum volume concentration of the PVSD. This was not correctly explained. We have changed this in the new version of the manuscript:*

*"when the maximum volume concentrations of the PVSD oscillate from 0.01 to 0.04"*

**RC: Line 269: How trustworthy are the SSA values? The MOSAiC multiwavelength lidar observations also show smoke SSA values of 0.95-0.96 (Ohneiser et al., 2021). Could be mentioned as a support.**

*AC: As it was indicated in line 112 of the old version of the manuscript, we only use AERONET retrievals which present a sky error lower than 10%, which is an indicator of the quality of the retrievals. Therefore the SSA values are trustworthy.*

*This has been included in the new version of the manuscript:*

*"These retrieved properties have been filtered by the residual obtained by the inversion algorithm; inversions with residuals bigger than 10 \% have been rejected, which ensures the quality of the retrievals."*

**RC: Table 2: The figure caption should mention that the products are derived from photometer observations**.

*AC: This has been corrected in the new version of the manuscript.*

**RC: Line 275: What do you expect from the surface observations? Washout processes continuously remove particles entering the PBL from above. Smoke plumes travel in the free troposphere and in the ULTS height region. Raikoke aerosol travels at stratospheric heights. So, what can tell us the Arctic surface observations about smoke and volcanic aerosols in the Arctic?**

*AC: This work focuses on the Exceptional high AOD observed over Svalbard in summer 2019 from a multi-instrumental perspective, we also intend to compare how the in-situ correlates with the columnar data. Of course, due to the high altitudes at which some events are arriving to Svalbard we usually do not see a correlation with the in-situ data. Nevertheless, it can be seen that in the first episode there is a correlation. This analysis of both columnar and surface indicates that AOD data is not usually representative of surface conditions and yet, AOD data misses some surface pollution events.*

*This has been included in the conclusions in the new version of the manuscript to make clearer the use of the surface observations.*

*"Therefore, column-integrated measurements are not representative of surface conditions, and they may miss some surface pollution events."*

**RC: Line 316 (page 14): Stratospheric AOD values as presented in Figure 6 should also be discussed in the main text body and contrasted to the volcanic AODs. The volcanic AODs are probably at all smaller than 0.025 with occasional exceptions, but at all below 0.05.**

*AC: Unfortunately, the CALIOP vertical feature mask layer classification in the stratosphere alone does not allow us the clear distinguishing between volcanic and smoke layers. Both, sulfate and smoke layer are observed during the time period and also a larger number of unclassified layers are observed. We have added the layer classification to the supplement (below a color-coded overview Figure).*

[Figure]

**RC: Line 321: Only four KARL (ground-based lidar) observations in 4 months (120 days)! This is really bad news!**

*AC: We already explained the reason for this in a previous comment.*

*We have included a remark in the manuscript: "Unfortunately, KARL measurements are only available for four days in the summer of 2019 due to cloudy conditions and safety regulations."*

**RC: Figure 6: This figure showing CALIPSO profiles is also of low quality. Many height bins are zero! How can we trust such extinction profiles? What are the numbers for the extinction peaks? There is no x-axis for extinction values. Again, what particle lidar ratio is used in the CALIOP data analysis? Many peaks are above the tropopause, some are caused by smoke (then a lidar ratio of 70 sr is appropriate) and some by volcanic sulfate layers (then a lidar ratio of 40 sr should be used in the Fernald retrieval). Any comment on this issue is welcome!**

*AC: This is due to the layer-based retrieval, thus only where aerosol layer was identified, the extinction is given. This "zero" means no distinguished layer was found in this altitude region. We realize that this can be a bit confusing. Thus, we modified the figure caption to make this clearer and we also have included an x-axis scale for extinction values:*

*"Time series of the extinction profiles at 532 nm measured by CALIOP in the summer of 2019. For each time period where aerosol layer was identified, the enhancement of the extinction within the layer is shown; the zero line indicated the date-time of observations. For reference, an x-scale for the extinction profiles has been included in green in the first profile The blue lines indicate the tropopause. The tropospheric and stratospheric AOD at 532 corresponding to each profile is included in the upper panel; the corresponding uncertainty of the AOD is given by the bars. The red shaded areas indicate the days on which the columnar events were identified (CS1, C2 and C3)."*

*We generally find one tropospheric layer and up to three stratospheric layer, see Figure S3 in the Supplement. To convert the measured backscatter to estimated extinction, lidar ratios, as described in Kim et al. (2018) and Tackett, et al., (2023) are used by the CALIOP teams. For the stratosphere these are 50 sr ± 18 sr for sulfate and unclassified layers, and 70 sr ± 16 sr for smoke.*

*As written above, uncertainties can be large, therefore we have added given uncertainties to the tropospheric and stratospheric AOD estimates.*

**RC: Figure 7: What is shown? Is the backscatter signal profile shown? The color scale indicates: the ratio of the aerosol backscatter to the Rayleigh backscatter coefficient is shown. Please clarify!**

*AC: According to this and the rest of your comments about these plots we have we exchanged them, see below:*

[Figure]

*The old figure has been moved to the supplement material to show the temporal stability.*

*The corresponding discussion has been updated in the new version of the manuscript:*

*"Unfortunately, KARL measurements are only available for four days in the summer of 2019 due to cloudy conditions and safety regulations of the instrument. The layers observed with KARL were temporally quite constant on each day (See Figure S4 in the supplement), therefore, the daily averaged backscatter profiles have been calculated. These are shown in Figure 7. It is observed and increased backscatter between 10 and 16 km a.g.l. with several layers through August, but in 17 September the backscatter slowly decreases and becomes more homogenous with height. Only one of these days corresponds to a day identified with aerosol event, 11 August (Event C3). During this day, a high backscatter coefficient at 532 nm up to about 0.8 M m − 1, with several layers, is observed throughout the entire troposphere, as well as in the stratosphere up to nearly 16 km a.g.l.. Particularly, the layer just around the around 10 km a.g.l. observed with KARL correlates very well in altitude with the increased backscatter profile measured by CALIOP in the same date In CALIOP it is also observed some extinction around 14 km a.g.l., which correlates with the stratospheric layers observed with KARL. Since the vertical and temporal resolution from both instruments is very different, we do not expect a closer agreement."*

**RC: I do not see any consistency between Figure 6 (always sharp layers with the vertical thickness of less than 1 km) and Figure 7 (vertically deep layers, partly 5-7 km thick, most layers without sharp edges).**

*AC: As vertical and temporal resolution are very different we do not expect a closer agreement. Basically, a ground-based lidar perceives the temporal evolution, a quick satellite the spatial evolution of aerosol layers.*

*This has been updated in the new version of the manuscript:*

*"Particularly, the layer just around the tropopause observed with KARL correlates very well in altitude with the increased backscatter profile measured by CALIOP at around 10 km a.g.l. in the same date, considering that vertical and temporal resolution from both instruments is very different."*

**RC: Why do you not show any figure with all the basic lidar profiles, i.e., height profiles of the backscatter coefficient an 355, 532, and 1064 nm together with the particle depolarization ratio. This would be a convincing figure as an introduction to Figure 8!**

*AC: We exchanged Fig 7 and now we show the daily averaged backscatter profiles. The depolarization is more boring, close to 3 % with the exception of clouds. We do not show this for clarity of the plots. In the following example you can appreciate it:*

[Figure]

**RC: Figure 8 shows some kind of inversion products (effective radius estimates). How can I trust Figure 8? Without showing any figure with the input profiles for the effective radius retrieval, I have to conclude that these backscatter profiles are of rather low quality so that the retrieved effective radius values are also of rather low quality (and therefore not trustworthy).**

*AC: Mistrust in science is a good thing! However, neither your remark on poor data quality nor a strong impact of the chosen LR is true.*

**RC: As mentioned above: Each backscatter computation at 355, 532, and 1064 nm needs quite very different lidar ratios as input! And these lidar ratio spectra are quite different for smoke and for sulfate. The input lidar ratio sets have a big impact on the quality of the determined spectrum of the backscatter coefficients, and consequently, on the effective radius values. Again: How trustworthy are the computed backscatter ratios and at the end the estimated effective radius values? All this must be discussed in the paper.**

*AC: As the Raman-Lidar is only one of many instruments (with poor data coverage) we extend the discussion a bit but still keep it short. Neither the choice of a LR (e.g. by the shadow or transition method of Chen et al [https://doi.org/10.1364/AO.41.006470](https://doi.org/10.1364/AO.41.006470)) nor a lengthy discussion on the uncertainty of d beta / d LR is aim of this work. However, you are right that our initial description of the data evaluation in 2.2.3 is too short. We added:*

*"The aerosol backscatter profiles at the three available wavelengths have been calculated with 60 m and 600 s resolution according to Klett (Speidel and Vogelmann, 2023) with clear sky approximation (aerosol backscatter small against molecular backscatter at altitudes > 22 km) and the choice of a prescribed lidar ratio (Ritter and Münkel, 2021) of 70, 45 and 45 sr for the wavelengths of 355, 532 and 1064 nm, respectively. The lidar ratios for 355 and 532 nm have been verified by backscatter values in the clear troposphere. An uncertainty of ±10sr for the lidar ratio has been estimated, giving rise to about 10 % uncertainty in the derived aerosol backscatter. For the 1064 nm wavelength the uncertainty is dominated by the assumed backscatter >22km as a boundary condition, such that also 10% uncertainty at this wavelength is realistic. Data points in time and altitude which were covered by clouds have been removed to not bias aerosol properties."*

**RC: Ohneiser et al. (ACP; 2021) show effective radii for the Siberian smoke of 0.2-0.22 micrometer. Since the size distribution of volcanic sulfate particles is similar (well-defined accumulation mode) similar effective radii are expected for the Raikoke particles.**

*AC: Ohneiser's event is later in the season and purely stratospheric. If at all it is comparable to our Sep 17th event. With some scatter we also obtain reff >=0.2μm for the stratosphere for that day. So this fits together.*

**RC: Lines 332-355: The discussion on page 16 and 17 is very speculative. Speculations should be avoided as much as possible. The effective radius values shown Figure 8 for the troposphere are confusing. It is impossible to determine the effective radius for both the fine mode and for the coarse mode from just three backscatter coefficients.**

*AC: We refer again to line 150 of the old version of the manuscript. We do not assume bi-model distributions and we do not discuss them.*

*While it is true that speculation should be avoided, it is clear from Fig. 8 that larger particles are found in the troposphere and hygroscopic growth is the main reason for this. If you have further suggestions how to interpret the Figure 8 we would like to discuss it with you.*

**RC: Line 365: A discussion on self-lofting of smoke layers needs to be included. As mentioned aged smoke particles (i.e., smoke older than 2-3 days) consists to 2-3% of BC and 97% of OC. For these BC-OC particles, lofting efficiencies must be estimated (Ohneiser et al., ACP 2023). One may show Figure 9, but needs to discuss the potential impact of lofting that shifts the profiles upward, towards greater heights. In this discussion, the findings of Ohneiser et al. (ACP, 2021), partly summarized in Ansmann et al. (JGR, 2024), maybe helpful. The MOSAiC observations show the aerosol pollution conditions in the Arctic for October-November 2019. Self-lofting probably came to an end in September 2019, before the MOSAiC expedition started.**

*AC: FLEXPART co-authors experts consider that self-lofting calculation in FLEXPART is not feasible. However, we have included the following discussion in the results, Section 3.2.1:*

*"As mentioned in the introduction, during the MOSAiC expedition a persistent 10 km deep aerosol layer in the UTLS, roughly from 7-8 km up to 17-18 km over the central Arctic, with clear a sign of smoke was observed. A layer around 10-15 km has also been observed in the data for summer 2019 analyzed here. Therefore, some lifting of the smoke must have taken place. The air in July-August 2019 originated from ongoing large wildfires over Siberia and low-wind and stagnant conditions allowed air to accumulate. The lack of evidence of strong pyrocumulonimbus (pyroCb) activity over these fires during the key period in combination with CALIPSO smoke detections at 10 km, led Ohneiser et al. (2021) to invoke that self-lofting might be a possible mechanism resulting in the persistent UTLS smoke layer. In a more recent publication, Ohneiser et al. (2023) explicitly treat self-lofting as a credible alternative to pyroCb convection for raising large smoke masses from 2-6 km to the tropopause and cites the 2019 Siberian case and MOSAiC results as key evidence. In addition, (Tarshish and Romps, 2022) tried to answer whether a dry firestorm plume (an intense conflagration that creates and sustains its own updraught wind system) can on its own reach the stratosphere. By using plume models (with and without entrainment), direct numerical simulations (DNS) and large-eddy simulations (LES) of idealized urban firestorms, they found that a dry plume starting at around 1 km (top of PBL) needs a temperature anomaly of about 60 K to stay positively buoyant up to a 15 km tropical tropopause. When they included entrainment, they found that for 1 km plume radius, mixing doubles temperature anomaly in the poles and sextuples it in the tropics. They conclude that narrow and dry plumes need to be unrealistically hot to reach stratospheric heights. Then, they used DNS and LES to simulate realistic dry firestorms and found that they never get hot enough to reach the stratosphere staying at around 5 km, at maximum. When relative humidity in the plume increased above 50 %, pyroCb-like convection developed, which lifted fire plumes to tropopause or even to stratosphere. They conclude that even moderately moist*

*environments allowed latent heating to push firestorm plumes to the stratosphere. Overall, whether the lifting of smoke in summer 2019 was due to pyroCb-like latent heating (moist convection) (Tarshish and Romps, 2022) or due to radiative heating (self-lofting) (Ohneiser et al., 2021) requires further research. While plume-rise parametrizations with moist thermodynamics and pyro-convection are already in use by many global models (Ma et al., 2024; Ke et al., 2025), they are not relevant here, as FLEXPART used emissions from CAMS GFAS."*

**RC: Lines 395-399: The maximum conversion of SO2 into sulfate occurs about 6 weeks after a volcanic eruption. For Raikoke (22 June eruption) the maximum sulfate load should have been observed around 10 August. However, freshly formed volcanic aerosol layers are usually organized in sharp layers in the stratosphere as indicated by the CALIOP observations in Figure 6. The thick layer from 7 to 15 km is, to my opinion, a composite of smoke layering with sulfate layer contributions on top (in the stratosphere). Karl observations in Fig. 7 seem to be in line with this hypothesis. However, it cannot be excluded that some dense smoke layers also entered the lower stratosphere (by self-lofting). Thus, the interpretation of the observations needs to be carefully done. The observations in Europe (Vaughan et al., 2021) have to be handled with caution as well. It remains open to what extent smoke and sulfate contributed to the observed aerosol pollution in the stratosphere over the UK.**

*AC: We agree with the reviewer. It is still an open question to what extent smoke and sulfate contributed to the observed aerosol pollution over Europe. However, current knowledge indicates that probably the contribution of smoke was more important, more information is now given in the introduction:*

*"Antokhina et al. (2023) analyzed the large-scale features of atmospheric circulation to investigate the causes of the natural disasters happening in summer 2019 in Siberia. They found that a severe anticyclonic blocking in Siberia in summer 2019 led to pronounced forest fires in the northern part of Siberia and flooding in the eastern part. This high pressure system might have transported smoke aerosol first northwards into the Arctic and then eastwards towards the American sector."*

*and in the conclusions:*

*"The anticyclonic system observed in Siberia (Antokhina et al., 2023) likely enhanced the transport of aerosol to the Arctic, first northwards into the Arctic and then eastwards towards North America. Hence we may have seen the rest of this mixed smoke. These mechanisms suggest that the BB contribution was likely more important than the volcano contribution in the upper troposphere - lower stratosphere (UTSL)."*

**RC: The conclusion section as well as the Abstract need to be updated after the revision of the main parts of the manuscript.**

*AC: The abstract and conclusions have been updated in the new version of the manuscript according to the referee comments and updates on the manuscript.*

**RC: Lines 493-500: Dedicated (future) field campaigns make sense, but all the methods, techniques and instruments are already available since decades, but well designed actions have never been conducted in the Arctic.**

*AC: Thanks for the remark, this has been included in the new version of the manuscript.*

*"It is obvious that there is a strong need for dedicated campaigns to bring together all methods of AOD studies, including both the in situ and remote sensing ones. While methods, techniques and instruments are already available since decades, well designed actions have never been conducted in the Arctic."*

**RC: Figure 9: AFR in the panel, AF in the caption.**

*AC: This has been corrected in the new version of the manuscript.*

---

## Author Comment (AC3)

**Response to the Anonymous Referee #2 comments for the manuscript "Exceptional high AOD over Svalbard in Summer 2019: A multi-instrumental approach" by Sara Herrero-Anta et al. in AMT**

*First of all, we would like to thank the time and effort of the referee for their detailed review of the manuscript. Reviewer comments (RC) are in bold font and author comments (AC) are in italic font.*

Author's answer to Anonymous Referee #2

**RC: The paper "Exceptional high AOD over Svalbard in Summer 2019: A multi-instrumental approach" by Herrero-Anta et al. is a thorough study of an episode of enhanced aerosol at Svalbard during summer 2019. It combines observations of several instruments to identify aerosol characteristics. Further, modelling with FLEXPART is used to identify the different sources of the aerosol.**

**Overall, the paper presents a comprehensive, logically structured study, it is well written, and is of interest to the broad readership of ACP.**

**The paper is therefore recommended for publication in ACP after addressing my minor comments as detailed below.**

**My main comment is that in some places discussion of some features in the data is missing.**

**AC: SPECIFIC (MINOR) COMMENTS:**

**RC: (1) Fig.2: Please comment! Why is there a peak of AOD with strong standard deviation in the reference record between 1 and 15 July? Is this a repetitive event each year? Or is this peak attributable to a specific event? If yes, which?**

*AC: Yes, this is due to an extreme event observed on 10 and 11 July 2015, when AOD (500 nm) up to 1.0 were measured.*

*This has been included in the capture of Figure 2 in new version of the manuscript: "The high reference values observed on 10 and 11 July are due to an extreme event that occurred in 2015.*

**RC: (2) Fig.4: For the S1 event, beta_sca is only enhanced at GAL, but not at ZEP, while beta_abs is enhanced at both sites. Do you have any explanation for this?**

*AC: It is not that they Bsca was not enhanced, but that, unfortunately we did not record Bsca at ZEP exactly at the time. We also do not have data from GAL on event S2 and CS1.*

**RC: (3) l.301-311, about Fig.5: ZEP discussion is missing! Here you should also comment about the PNSD at ZEP, which does not show a clear bimodal structure during the surface events, and the distributions peak at sizes between Aitken and Accumulation mode.**

*AC: Thank you for the remark. We have included a short discussion in the new version of the manuscript:*

*"The PNSD at ZEP does not show a clear bimodal structure for either of the events. In these cases, the peak of the distribution lies between the Aitken and Accumulation modes, generally showing higher concentrations than GAL during events S1 and S2. Therefore, the surface events were perceived slightly differently at the two sites."*

**RC: (4) l.324-325: What about the other layers seen by KARL? There is a more intense layer at 13km that is not seen by CALIOP. Why? Is this an issue of the CALIOP sensitivity?**

*AC: We did not mentioned it but actually CALIOP also sees some extinction around 13-14~km in most of the profiles. As vertical and temporal resolution of CALIOP and KARL are very different we do not expect a closer agreement. Basically, a ground-based lidar perceives the temporal evolution, a quick satellite the spatial evolution of aerosol layers. In the new version of the manuscript we also have updated the discussion and Figures following other reviewer comments:*

*"Unfortunately KARL measurements are only available for four days in the summer of 2019 due to cloudy conditions and safety regulations of the instrument. The layers observed with KARL were temporally quite constant on each day (See Figure S4 in the supplement), therefore, the daily averaged backscatter profiles have been calculated. These are shown in Figure 7. It is observed and increased backscatter between 10 and 16 km a.g.l. with several layers through August, but in 17 September the backscatter slowly decreases and becomes more homogenous with height. Only one of these days corresponds to a day identified with aerosol event, 11 August (Event C3). During this day, a high backscatter coefficient at 532 nm up to about 0.8 $M\,m - 1$, with several layers, is observed throughout the entire troposphere, as well as in the stratosphere up to nearly 16 km a.g.l.. Particularly, the layer just around the around 10 km a.g.l. observed with KARL correlates very well in altitude with the increased backscatter profile measured by CALIOP in the same date In CALIOP it is also observed some extinction around 14 km a.g.l., which correlates with the stratospheric layers observed with KARL. Since the vertical and temporal resolution from both instruments is very different, we do not expect a closer agreement."*

**RC: (5) l.447-449: Thin cirrus clouds are hard to detect by ground based and space based instrumentation. Could thin (subvisible) cirrus clouds also contribute to negative values of delta-DNI?**

*AC: Optically very thin clouds can be difficult to detect even from ground-based sun-photometers, so this is a complex problem not only for $\Delta DNI$. Thin clouds (but also not so thin clouds) can have contributed to the negative values and to the variability of $\Delta DNI$.*

*However, the tendency observed from July to August and September must be due to the aerosol presence. In particular, the first event correlates very well with the moment at which we start to observe only negative ΔDNI.*

*This effects of clouds were already mentioned in the manuscript: "The large standard deviation observed shows the complexity of this analysis, with multiple conditions (mainly variation in aerosols and clouds) sometimes playing roles in opposite directions. However, in general, the negative sign of ΔDNI is a good proxy for the effect of the decrease in the direct component of solar radiation."*

**RC: TECHNICAL COMMENTS:**

**(1) l.28: levels.Lisok -> levels. Lisok**

**(2) l.52: Siberian wildfires -> smoke of the Siberian wildfires**

**(3) Table 1: abbreviations of several parameters (e.g., DNI, DIF) are only given later in the text. This should be mentioned in the table caption.**

**(4) l.148: can be also be -> can also be**

**(5) l.161: to Reference Upper-Air Network (GRUAN) -> to the Global Climate Observing System (GCOS) Reference Upper-Air Network (GRUAN).**

**(6) l.171: from zero to the unit, being small -> from zero to unity, with small**

**(7) caption of Fig.2, l.2-3: Sentence "Long-term daily means ..." can be deleted because same info is given at the end of the caption.**

**(8) caption of Fig.2, l.5: with errors bar -> with error bars**

**(9) l.253: one maxima -> one maximum**

**(10), (11) l.254: main maxima -> main maximum second maxima -> second maximum**

**(12) l.259: This longer radii -> These larger radii**

**(13) p.11, last line: longer -> larger**

**(14) l.277: next mean values: -> following mean values:**

**(15) Table 3 and text on p.13: Here you use B_abs and B_sca instead of beta_abs and beta_sca. Please use consistent notation throughout!**

**(16) caption of Fig.5: the are only -> there are only**

**(17) l.303: With respect GAL observations -> Regarding GAL observations**

**(18) l.331: is shown -> are shown**

**(19) l.361: row)is -> row) is**

**(20) Caption of Fig.10: red line -> magenta line**

**(21) Caption of Fig.11: With respect the sources -> With respect to the sources**

**(22) Caption of Fig.12: with respect the reference -> with respect to the reference**

**(23) l.495: Ship born -> Ship borne**

**(24) l.503: under request to the authors -> under request to the authors.**

*AC: Thanks for the detailed review, all the technical comments have been addressed in the new version of the manuscript.*

---

## Author Response (AR2)

Dear Editor Hinrich Grothe,

Thank you for your supervision. The technical corrections indicated by Anonymous referee #2 have been addressed in the final version of the manuscript.

Regarding the comment about the quality of the lidar backscatter profiles, we already indicated the uncertainty of the aerosol backscatter profiles in lines 164-165 of the manuscript: "An uncertainty of 10 sr for the lidar ratio has been estimated, giving rise to about 10% uncertainty in the derived aerosol backscatter."

Despite multiple aerosol layers, the uncertainty in 18 km altitude is about 10%, which is a satisfactory result. As it can be seen in the following figure, our derivation of the layer properties is not affected by noise; instead, the seven profiles shown in blue (10 min each) exhibit some temporal variability.

[Figure]

We hope this addresses your comment. If any issues remain unresolved or if you have further comments, please let us know.

Best regards,

Sara Herrero-Anta et al.